# Anti-PD-L1-Based Bispecific Antibodies Targeting Co-Inhibitory and Co-Stimulatory Molecules for Cancer Immunotherapy

**DOI:** 10.3390/molecules29020454

**Published:** 2024-01-17

**Authors:** Qiaohong Geng, Peifu Jiao

**Affiliations:** School of Chemistry and Chemical Engineering, Qilu Normal University, Jinan 250200, China

**Keywords:** co-inhibitory molecules, co-stimulatory molecules, anti-PD-L1-based bsAbs, cancer immunotherapy

## Abstract

Targeting PD-L1 via monospecific antibodies has shown durable clinical benefits and long-term remissions where patients exhibit no clinical cancer signs for many years after treatment. However, the durable clinical benefits and long-term remissions by anti-PD-L1 monotherapy have been limited to a small fraction of patients with certain cancer types. Targeting PD-L1 via bispecific antibodies (referred to as anti-PD-L1-based bsAbs) which can simultaneously bind to both co-inhibitory and co-stimulatory molecules may increase the durable antitumor responses in patients who would not benefit from PD-L1 monotherapy. A growing number of anti-PD-L1-based bsAbs have been developed to fight against this deadly disease. This review summarizes recent advances of anti-PD-L1-based bsAbs for cancer immunotherapy in patents and literatures, and discusses their anti-tumor efficacies in vitro and in vivo. Over 50 anti-PD-L1-based bsAbs targeting both co-inhibitory and co-stimulatory molecules have been investigated in biological testing or in clinical trials since 2017. At least eleven proteins, such as CTLA-4, LAG-3, PD-1, PD-L2, TIM-3, TIGIT, CD28, CD27, OX40, CD137, and ICOS, are involved in these investigations. Twenty-two anti-PD-L1-based bsAbs are being evaluated to treat various advanced cancers in clinical trials, wherein the indications include NSCLC, SNSCLC, SCLC, PDA, MBNHL, SCCHN, UC, EC, TNBC, CC, and some other malignancies. The released data from clinical trials indicated that most of the anti-PD-L1-based bsAbs were well-tolerated and showed promising antitumor efficacy in patients with advanced solid tumors. However, since the approved and investigational bsAbs have shown much more significant adverse reactions compared to PD-L1 monospecific antibodies, anti-PD-L1-based bsAbs may be further optimized via molecular structure modification to avoid or reduce these adverse reactions.

## 1. Introduction

Cancer immunotherapy has been experiencing a renaissance since the first regulatory approval of the anti-CTLA-4 antibody ipilimumab in 2011 by the FDA [1,2,3]. After that, a growing number of monospecific antibodies targeting both co-inhibitory and co-stimulatory molecules have been developed and approved [4,5,6]. The main idea of cancer immunotherapy is to enable immune cells such as T cells to recognize and destroy cancer cells [7,8,9]. By now, cancer immunotherapy has joined the ranks of surgery, radiotherapy, chemotherapy, and targeted therapy as a central pillar to conquer this deadly disease [10,11,12,13,14,15].

Programmed cell death 1 ligand-1 (PD-L1, also known as CD274 and B7-H1), identified by Lieping Chen in 1999, represents one of the most important co-inhibitory molecules for cancer immunotherapy [16]. PD-L1 is a type I transmembrane protein containing 290 amino acids encoded by the *Cd274* gene on human chromosome 9 [17]. The wild-type PD-L1 is composed of a signal peptide, an extracellular IgV and IgC domain, a hydrophobic transmembrane region, and a short cytoplasmic tail [18]. PD-L1 protein is broadly expressed on the surface of specimens that are freshly isolated from patients with various cancers, while the normal human tissues do not express PD-L1 protein except for cells from the macrophage lineage. This unique expression profile was explained by the finding that IFN-γ upregulates PD-L1 on the surface of various cancer cells [19]. Recent advances indicated PD-L1 plays important roles in the regulation of the inflammatory process in cancer cells [20,21] and immune cells [22], and the regulation of genomic stability in cancer cells [23,24,25,26]. The engagement of PD-L1 with one of its receptors, PD-1, leads to the negative regulation of CD3-mediated lymphocyte proliferation [27] and the deletion of activated lymphocytic T cells in the tumor microenvironment (TME) [19]. PD-L1 also engages with another receptor, CD80, to limit T-cell co-stimulation [28]. As a result, the expression of PD-L1 in the TME on the surface of cancer cells plays central roles in tumor-induced immune escape [18]. The blockade of PD-1/PD-L1 or CD80/PD-L1 interactions using PD-L1 inhibitors such as antibodies, peptides, and small molecules represents a promising therapeutic strategy for cancer immunotherapy [29,30,31,32,33,34,35,36].

Targeting PD-L1 via monospecific antibodies has shown durable clinical benefits and long-term remissions where patients exhibit no clinical cancer signs for many years after treatment [37,38,39,40,41]. Six monospecific antibodies (Table 1) have received their regulatory approval for the treatment of urothelial carcinoma (UC), Merkel cell carcinoma (MCC), renal cell carcinoma (RCC), colorectal cancer (CC), non-small-cell lung cancer (NSCLC), small-cell lung cancer (SCLC), hepatocellular carcinoma (HCC), biliary tract cancer (BTC), metastatic triple-negative breast cancer (mTNBC), melanoma, and sarcoma. Most common adverse reactions of PD-L1 monospecific antibodies in patients (≥20%) are fatigue, decreased appetite, nausea, urinary tract infection, pyrexia, constipation, dyspnea, cough, and musculoskeletal pain. However, the durable clinical benefits and long-term remissions by anti-PD-L1 monospecific antibodies have been limited to a small fraction of patients (around 20%) with certain cancer types [42,43]. Targeting PD-L1 via bispecific antibodies which can simultaneously bind to both co-inhibitory and co-stimulatory molecules may increase durable antitumor responses in patients who would not benefit from monospecific antibodies.

BsAb was first described by Nisonoff and co-workers more than six decades ago [44] and has provided exciting opportunities for cancer immunotherapy. As of 2023, thirteen bsAbs have received their regulatory approvals, and eleven of them are developed for the treatment of various cancers including solid tumors and hematologic malignancies (Table 2). BsAb represents a diverse family of proteins designed to recognize two different epitopes or antigens. There are two major formats of bsAbs that range from relatively small proteins (non-IgG-like) to large molecules (IgG-like) with additional domains attached (Figure 1) [45]. Current anti-PD-L1-based bsAbs are exclusively IgG-like formats. The IgG-like bsAbs have antibody-dependent cell-mediated cytotoxicity (ADCC), complement dependent cytotoxicity (CDC), and antibody-dependent cellular phagocytosis (ADCP) effects and a long half-life time due to the presence of Fc domain. However, non-IgG-like bsAbs such as F(ab)_2_, Tandem scFv (BiTEs), Tandem V_HH_s, and scFv-Fab have a relatively short plasma half-life, as they lack protection from catabolism by the neonatal Fc receptor, FcRn. An attractive feature of bsAb is their temporal or spatial linkage of the two binding specificities (cells or proteins) that is not present in any combination of parent antibodies. This linkage can recruit various therapeutic immune cells in tumor tissues and provide exciting opportunities for anti-PD-L1 therapy [46]. Since PD-L1 is mainly expressed on the surface of tumor cells, anti-PD-L1-based bsAbs may recruit various therapeutic immune cells in TME and exerts synergistic antitumor effects, thus opening a new window for cancer immunotherapy. So far, a growing number of anti-PD-L1-based bsAbs simultaneously targeting both co-inhibitory and co-stimulatory molecules have been investigated in biological testing or in clinical trials (Figure 2 and Figure 3 and Appendix A).

## 2. Anti-PD-L1-Based bsAbs Targeting Co-Inhibitory Molecules

### 2.1. CTLA-4 × PD-L1

Cytotoxic T lymphocyte-associated antigen 4 (CTLA-4, also known as CD152), identified by Brunet JF in 1987, is the first co-inhibitory molecules for cancer immunotherapy. CTLA-4 contains an IgV domain, a transmembrane domain, and a cytoplasmic tail [47]. CTLA-4 is expressed on the surface of CD4^+^ helper and CD8^+^ effector T cells (Teffs), as well as CD4^+^CD25^+^Foxp3^+^ regulatory T cells (Tregs) in the process of T-cell activation [48]. The B7 family proteins of CD80 (B7-1) and CD86 (B7-2) are expressed on antigen-presenting cells (APCs) and T cells, and provide a major co-stimulatory signal for the T-cell response via interaction with CD28 on T cells [49]. CD28 is expressed constitutively on T cells while CTLA-4 is rapidly upregulated after T-cell activation within the secondary lymphoid organs. CTLA-4 shows a much higher affinity (10~20-fold) to co-stimulatory CD80 and CD86, and eventually competes with CD28 to function as a negative regulator for T-cell activation [50]. If the task of tumor eradicating has not been completed whereas the expression of CTLA-4 is triggered, the T cells will be inactivated and unable to complete this complex task. As a result, anti-CTLA-4 therapy using antibodies allows for long-term effective immune responses [51]. Since the CTLA-4 pathway functions in secondary lymphoid organs while the PD-L1 pathway works within the TME, targeting CTLA-4 and PD-L1 simultaneously using bsAbs may enhance the antitumor benefit for cancer immunotherapy.

Researchers from Nanjing Legend Biotech disclosed an IgG-like bsAb (BCP-84 and BCP-85) simultaneously targeting CTLA-4 and PD-L1 in 2018. BCP-84 and BCP-85 each consist of two copies of the first polypeptide and two copies of the second polypeptide. The Fc regions were constructed from non-glycosylated IgG1 with a S228P mutation and had no binding affinity to FcγRs, thus avoiding the depletion of the PD-L1- or CTLA-4-positive cell by ADCC. BCP-84 and BCP-85 showed higher tumor inhibition efficacy over either of anti-PD-L1 or anti-CTLA-4 monotherapy in C57BL/6CTLA-4 knock-in mice bearing a colorectal MC38 tumor. Notably, the anti-tumor efficacy of BCP-84 or BCP-85 was comparable to the combination therapy [52].

Biologists from Sichuan Kelun-Biotech Biopharmaceutical disclosed an IgG-like bsAb (AB-04) simultaneously targeting CTLA-4 and PD-L1 in 2019. AB-04 is composed of an IgG domain specifically binding to PD-L1 and a scFv domain specifically binding to CTLA-4. AB-04 had ADCC and CDC activities. AB-04 significantly increased IL-2 secretion in superantigen (SEB)-stimulated PBMCs compared with its parental antibody. AB-04 showed significant anti-tumor efficacy at a concentration of 10 mg/kg with a tumor growth inhibition (TGI) value of 67.72% in NSG mice which were subcutaneously transplanted with human non-small-cell lung cancer HCC827 cells. Toxicological study indicated that AB-04 decreased the toxicity of the CTLA-4 antibody and improved the safety of the drug to some extent [53].

Scientists from Jiangsu Alphamab Biopharmaceuticals disclosed an IgG-like bsAb (KN-046, Erfonrilimab) simultaneously in 2019. KN-046 belongs to the Tandem V_HH_-Fc format bsAb wherein immunoglobulin single variable domains (ISVDs) specifically bind to PD-L1 and CTLA4, respectively. KN-046 significantly increased IL-2 secretion in SEB-stimulated PBMCs at the concentration of 15 nM. KN-046 showed significant anti-tumor efficacy at doses of 0.3 mg/kg or above in the A375-hPD-Ll/PBMC xenograft mouse model and the MC38-hPD-L1/Double KNOCK-IN model. KN-046 was more quickly enriched in the tumors, indicating better safety and less toxicity compared with the combination of the anti-PD-1 antibody and anti-CTLA4 antibody. The phase I results in patients with advanced solid tumors revealed that KN-046 was well-tolerated and showed promising antitumor efficacy in advanced solid tumors, especially in patients with nasopharyngeal carcinoma. The most common treatment-related adverse events (TRAEs) were rash (33.0%), pruritus (31.0%), and fatigue (20.0%). Grade ≥ 3 TRAEs were observed in 14.0% of participants. The phase II results in patients with non-small-cell lung cancer who failed platinum-based chemotherapy showed that KN-046 had promising efficacy and a good safety profile for advanced NSCLC after failure or intolerance to previous platinum-based chemotherapy. Blinded independent review committee (BIRC)-assessed objective response rates (ORRs) were 13.3% and 14.7% in the 3 mg/kg and 5 mg/kg cohorts, respectively. Median progression-free survival was 3.68 and 3.68 months, while overall survival was 19.70 and 13.04 months, respectively. The most common TRAEs were anemia (28.1%), hyperglycemia (26.7%), and infusion-related reactions (26.7%). The incidence rates of grade ≥ 3 TRAEs and TRAEs leading to treatment discontinuation were 42.2% and 14.1%, respectively [54,55,56]. Two phase III studies in patients with advanced squamous NSCLC and advanced pancreatic ductal adenocarcinoma were initiated in 2021 and 2023, respectively.

Biologists from Harbour BioMed disclosed an IgG-like bsAb (PR-001573) in 2022. PR-001573 is a tetravalent symmetric bispecific antibody fusion (IgG-VH) construct comprising IgG-antibody-derived two Fabs (VH-VL-CH1-CL) targeting PD-L1, heavy-chain-antibody-derived two VH chains targeting CTLA4, and the IgG1λ Fc (CH2-CH3) domain. PR-001573 was among the bispecific antibody fusion constructs in this patent. In vivo, the administration of PR-001573 significantly inhibited tumor growth with a TGI value of 100% and did not change the weight of hPD-L1-expressing colon cancer (MC38)-cell-grafted PD1- and CTLA4-transgenic female B6 mice [57].

### 2.2. LAG-3 × PD-L1

Lymphocyte Activation Gene 3 (LAG-3, also known as CD223), which was cloned in the early 1990 as a CD4 homologue, is a type I transmembrane protein with four extracellular Ig-like domains (D1–D4), and a cytoplasmic region responsible for LAG-3 signaling [58]. The cytoplasmic region has an EP (glutamic acid/proline) motif that associates with LAG-3-associated protein, as well as a KIEELE motif thought to be required for the LAG-3 modulation of T-cell function [59]. Like CD4, LAG-3 binds to MHC class II molecules but with a higher affinity than CD4 [60]. LAG-3 is expressed on activated T cells, NK cells, pDCs, B cells, and γδT cells, and participates in immune suppression, particularly through persistent strong expression in a percentage of Tregs [61]. The immune-suppressive mechanism of LAG-3 on T cells is thought to be driven by the crosslinking of LAG-3 on activated T cells, resulting in decreased calcium flux and IL-2 release during T-cell activation [62].

Scientists from F-Star Delta Limited disclosed an IgG-like bsAb (FS-118) simultaneously targeting LAG-3 and PD-L1 in 2017. FS-118 is a bispecific IgG1 (148 KDa) antibody comprising a LAG-3 binding site located in a CH3 domain of the Fc region and a PD-L1 binding site in the Fab region. FS-118 enhanced the bridging between T cells and tumor cells via the dual targeting of LAG-3 and PD-L1 in TME. Phase I results in patients with advanced cancers indicated that FS-118 was well-tolerated with no dose-limiting toxicities (DLTs) observed up to and including 20 mg/kg quaque week (QW). The overall disease control rate (DCR) was 46.5% [63]. In 2017, F-star granted an option to Merck KGaA to in-license rights to develop and commercialize the antibody. However, in 2019, Merck KGaA declined to exercise this option. F-star retains exclusive rights to develop and commercialize FS-118.

Researchers from ABL Bio disclosed an IgG-like bsAb (ABL-501) simultaneously targeting LAG-3 and PD-L1 in 2020. ABL-501 was generated using the Grabody^TM^ platform and comprised an anti-PD-L1 antibody or Fab thereof and an anti-LAG-3 antibody or Fab thereof, wherein ABL-501 was capable of specifically binding to the immunoglobulin C domain of PD-L1 via at least one of amino acids Y134, K162, and N183. ABL-501 did not bind to the Ig V domain of PD-L1. ABL-501 enhanced the activation of CD4^+^ and CD8^+^ T cells with a higher degree compared with the combination of anti-LAG-3 and anti-PD-L1 antibodies. ABL-501 mitigated Treg-cell-mediated immunosuppression by augmented effector T-cell responses. Moreover, the binding of ABL-501 to LAG-3 and PD-L1 promoted dendritic cell (DC) activation and tumor cell conjugation with T cells that subsequently enhanced effective CD8^+^ T-cell responses. ABL-501 demonstrated its in vivo antitumor efficacy in a double-humanized xenograft model (hLAG-3/hPD-L1) [64].

Biologists from Innovent Biologics disclosed an IgG-like bsAb (IBI-323) simultaneously targeting LAG-3 and PD-L1 in 2020. IBI-323 comprised an anti-LAG-3 antibody and two anti-PD-L1 sdAbs. Anti-PD-L1 sdAbs were fused to the C-terminus of the heavy chain of the anti-LAG-3 antibody via a flexible linker of (GGGGS)_2_. The Fc-mediated ADCC and CDC effects were decreased by the introduction of LALA mutations (LALA, L234A, and L235A). IBI-323 could enhance T-cell activation by crosslinking PD-L1^+^ APCs with LAG-3^+^ T cells, and promote IFN-γ and IL-2 secretion in a dose-dependent manner. A dose-dependent tumor growth regression induced by IBI-323 was observed in human A375 melanoma and MC38 colon adenocarcinoma tumor xenograft model [65].

Scientists from GenScript Biotech Corp disclosed an IgG-like bsAb (mPD-L1HCv1-E-sLAG3) simultaneously targeting LAG-3 and PD-L1 in 2020. mPDL1HCv1-E-sLAG3 was a single domain antibody fused to a monoclonal antibody construct comprising a polypeptide chain consisting of an anti-PD-L1 heavy chain fused with an anti-LAG-3 single domain antibody (sdAb, V_HH_), and a second polypeptide chain consisting of an anti-PD-L1 light chain, expressed in CHO-3E7 cells. mPD-L1HCv1-E-sLAG3 inhibited the interaction between PD-1 and PD-L1 with high affinity (EC_50_ = 0.3291 nM), as determined by a PD-1/PD-L1 blockade bioassay. Moreover, mPD-L1HCv1-E-sLAG3 inhibited the interaction between LAG-3 and the major histocompatibility complex II (MHC II, inhibits T-cell signaling) with high affinity (EC_50_ = 13.86 nM), as determined by a LAG-3 blockade bioassay [66].

Biologists from WuXi Biologics disclosed an IgG-like bsAb (W-3669) simultaneously targeting LAG-3 and PD-L1 in 2021. W-3669 was a bispecific antibody consisting of humanized sdAb targeting hPD-L1 fused in tandem with human IgG1-Fc-domain-harboring L234A and L235A mutations, a sdAb targeting human LAG-3. W-3669 bound to recombinant hPD-L1 and LAG-3. W-3669 bound to hPD-L1 expressed on W315-CHO-K1.hpro1.C11 (EC_50_ = 0.20 nM) and LAG-3 on W339-FlpIn293 cells (EC_50_ = 0.74 nM). In addition, W-3669 exhibited dual binding affinity towards both PD-L1- and LAG-3-expressing cells. W-3669 blocked the binding of PD-1 to PD-L1 with an IC_50_ of 0.11 nM and LAG-3 to MHC-II with an IC_50_ of 1.65 nM. The incubation of W-3669 with PD-1- and LAG-3-expressing Jurkat cells in the presence of PD-L1^+^ APCs and Raji cells, separately, resulted in a significantly enhanced NFAT signaling pathway. Similarly, the co-culturing of PBMCs-derived CD4^+^ T cells and allogeneic immature dendritic cells (iDCs) with W-3669 resulted in increased IL-2 and IFN-γ secretion, indicating T-cell activation. In an in vivo assay, the administration of W-3669 to mouse colon carcinoma (Colon26)-induced syngeneic mice resulted in significantly increased tumor growth inhibition. In a pharmacokinetic assay, the half-life and area under concentration time curve (AUC) of W-3669 was found to be 133 h and 4928 h × mcg/mL in the serum of C57BL/6 mice [67].

Biologists from Mabwell (Shanghai) Bioscience disclosed an IgG-like bsAb (hz7F10-hzB6) simultaneously targeting LAG-3 and PD-L1 in 2022. Hz7F10-hzB6 comprised an anti-LAG-3 antibody and two anti-PD-L1 sdAbs. Anti-PD-L1 sdAbs were fused to the C-terminus of the heavy chain of the anti-LAG-3 antibody via a flexible Gly/Ser rich linker (GGGGSPGGGSPGGGS). The Fc-mediated ADCC and CDC effects were reduced by the introduction of LALA mutations (LALA, L234A, and L235A) and made the structure of the bispecific antibody more stable. Hz7F10-hzB6 showed significant anti-tumor efficacy with TGI values of 69% and 60% in MC38 and H1975 tumor xenograft models, respectively [68].

Researchers from Merus NV disclosed an IgG-like bsAb (PB-68) simultaneously targeting LAG-3 and PD-L1 in 2022. PB-68 exhibited excellent binding affinity towards human LAG-3 and PD-L1 as observed in a reporter assay (EC_50_ = 14.80 nM) and an SEB assay (EC_50_ = 0.34 nM). In vitro, the antigen-binding activity of PB-68 towards human LAG-3-transfected 293FF cells was observed to have the EC_50_ value of 0.27 nM. Similarly, the antigen-binding activity towards hPD-L1-transfected CHO-K1 cells was observed to have the EC_50_ value of 0.0097 nM. In a Staphylococcal enterotoxin D (ppSED) reporter assay, PB-68 displayed the EC_50_ value of 0.815 nM. In vivo, the administration of PB-68 to NSG mice grafted with MDA-MB-231 tumor cells reduced the tumor volume significantly [69].

### 2.3. PD-1 × PD-L1

Programmed cell death 1 (PD-1, also known as CD279), which was isolated in 1992, is a co-inhibitory molecule transiently expressed on the surface of activated immune cells such as T cells, B cells, NKs, DCs, and tumor-associated macrophages (TAMs) [27,70,71]. Sustained PD-1 expression is associated with T-cell dysfunction and tumor-induced immune escape in TME [72]. Human PD-1 is a type I transmembrane protein containing an immunoreceptor tyrosine-based inhibitory motif (ITIM) and immunoreceptor tyrosine-based switch motif (ITSM) [73]. Once engaged with one of its two ligands, PD-L1 or PD-L2, PD-1 becomes phosphorylated by Src kinases at its ITIM and ITSM, which then recruit the phosphatases of Src homology 2 (SH2) domain-containing phosphatase 2 (SHP2) and/or SHP1, and subsequently inhibits both TCR and CD28 signaling [74,75]. Bispecific targeting PD-1 and PD-L1 simultaneously may provide an alternative therapeutic approach with enhanced antitumor activity, compared with monospecific PD-1 or PD-L1 antibodies.

Researchers from Beijing Hanmi Pharm disclosed an IgG-like bsAb simultaneously targeting PD-1 and PD-L1 in 2018. Hanmi bsAb comprised a first Fab region capable of binding to PD-L1 and a second Fab region capable of binding to PD-1, wherein the bsAb comprised a first Fc chain and a second Fc chain linked by one or more interchain disulfide bonds. The Fc chain of Hanmi bsAb comprised five amino acid substitutions as follows: the positions of 366 and 399 on the first Fc chain, and the positions of 351, 407, and 409 on the second Fc chain. The reduced anti-PD-L1 and anti-PD-1 semi-anitibodies were mixed in an equimolar ratio and subjected to a re-assembly reaction at 4 °C for 24 h to provide Hanmi bsAb. Compared to monoclonal anti-PD-1 and anti-PD-L1 antibodies, Hanmi bsAb promoted the secretion of cytokines such as IL-2 and IFN-γ more significantly and showed stronger anti-cancer efficacy in vitro and in vivo [76].

Scientists from Eli Lilly and Innovent Biologics disclosed an IgG-like bsAb (LY-3434172, IBI-318) simultaneously targeting PD-1 and PD-L1 in 2019. LY-3434172 comprised a PD-1 monovalent arm derived from LY3342903 (US10316089) and a PD-L1 monovalent arm derived from LY3300054 (US10214586). To prevent Fc-mediated effector function, an Fc-effector-null backbone was introduced into the bsAb by mutagenesis (L234A, L235A, and D265S). LY3434172 could significantly block the interactions of PD-1/PD-L1, PD-1/PD-L2, as well as PD-L1/CD80. LY3434172 enabled the bridging of PD-1- and PD-L1-expressing cells and resulted in enhanced T-cell activation at lower concentrations relative to each parental antibody and their combination in vitro [77]. LY3434172 treatments led to notable and enhanced antitumor activity in several humanized tumor xenograft models. The enhanced potency of LY3434172 was related to the bridging of cells by the bispecific antibody as evidenced through the enhanced development of the immune synapses [78]. A phase I trial to evaluate the safety and tolerability of LY3434172 in participants with advanced solid tumors was completed in 2021. No further trials were initiated.

Biologists from Sunshine Guojian Pharmaceutical disclosed an IgG-like bsAb (609-Fab-PD-L1-IgG4) simultaneously targeting PD-1 and PD-L1 in 2021. The 609-Fab-PD-L1-IgG4 comprised two polypeptide chains (long heavy chains) and four common light chains with a molecular weight of 238 KDa; 609-Fab-PD-L1-IgG4 showed similar pharmacokinetic properties with a half-life of 361 h (15.0 days) to anti-PD-L1 mAb. The tumor inhibition rate of 609-Fab-PD-L1-IgG4 in NSG mice bearing an NCI-H292 tumor was 96.0% which was much higher than those of nivolumab (50.5%) and atezolizumab (84.4%). The 609-Fab-PD-L1-IgG4 required no Fc modification and has no mismatch problems [79,80].

Biologists from Sunshine Guojian Pharmaceutical also disclosed an IgG-like bsAb (anti-PD-1 × PD-L1 bsAb) simultaneously targeting PD-1 and PD-L1 in 2021. Anti-PD-1 × PD-L1 bsAb comprised two polypeptide chains (long heavy chains) and two light chains. The anti-PD-1 scFv from anti-PD-1 mAb were linked to anti-PD-L1 mAb via the linkers of (GGGGS)_n_ at the terminals of Fc region. Anti-PD-1 × PD-L1 bsAb showed similar pharmacokinetic properties with a half-life of 300 h (12.5 days) to 609-Fab-PD-L1-IgG4 [81]. The anti-cancer potential of anti-PD-1 × PD-L1 bsAb in vitro and in vivo was not yet disclosed.

### 2.4. PD-L2 × PD-L1

Programmed cell death 1 ligand-2 (PD-L2, also known as CD273 and B7-DC), which was identified in 2001, is a second ligand for PD-1, shares only approximately 40% identity with PD-L1, and inhibits T-cell activation [82]. PD-L2 is a type I transmembrane protein that contains an immunoglobulin (Ig)-like V-type domain and an Ig-like C2-type domain in its extracellular region. PD-L2 was expressed not only in tumor cells but also in immune cells, and its high expression has been proven to play an important role in tumorigenesis and immune escape [83]. PD-L2 binds to PD-1 with an approximately three-fold higher affinity than does PD-L1, and, like PD-L1, sends an inhibitory signal which attenuates T-cell function [84]. PD-L2 also binds to the repulsive guidance molecule b (RGMb) to mediate immunotolerance in the lung [85,86]. A relatively high co-expression of PD-L1 and PD-L2 has also been observed in a number of cancers such as classical Hodgkin’s lymphoma, primary mediastinal large B-cell lymphoma, T-cell lymphoma, triple-negative breast cancer, gastric carcinoma, melanoma, squamous carcinoma of the lung, head and neck, and cervix and vulva, bladder cancer, and hepatocellular carcinoma [83,87,88,89]. Co-targeting PD-L2 and PD-L1 using bsAbs may provide an alternative therapeutic approach, with enhanced antitumor activity, compared with monospecific PD-L1 antibodies.

Biologists from The University of Texas System and ImmunoGenesis disclosed an IgG-like bsAb (IMGS-001) simultaneously targeting PD-L2 and PD-L1 in 2019. IMGS-001 re-envisions the starting point for cold tumor treatment with a novel, dual-specific PD-L1/PD-L2 antibody designed with an effector function to kill the immunosuppressive cells in the TME. Rather than building on the fragile 5% response rate seen with PD-L1 inhibition in cold tumors, the biologists returned to the science to develop the foundation for a PD-1 pathway blockade. Early readouts suggest that IMGS-001 may increase the cold tumor monotherapy response rate >5-fold and provides a strong therapeutic foundation for carefully designed add-on therapies to further optimize cold tumor outcomes [90].

Scientists from Biotheus disclosed an IgG-like bsAb (Bi-201) simultaneously targeting PD-L2 and PD-L1 in 2021. Bi-201 was a bispecific fusion antibody comprising a nanobody targeting PD-L1 fused via GS linker to a nanobody targeting PD-L2, wherein human IgG1 harbors the LALA (L234A/L235A) mutation. In an in vitro assay, Bi-201 significantly bound to hPD-L1 and PD-L2 proteins with KD values of 4.81 × 10^−10^ and 4.45 × 10^−10^ M, respectively. Furthermore, Bi-201 significantly bound to hPD-L1 and hPD-L2 on the cell surface of CHO-hPD-L1 and CHO-hPD-L2 cells. In another in vitro assay, Bi-201 inhibited the binding of PD-L2 and PD-L1 to PD-1 and also blocked the PD-L1/PD-1 and PD-L2/PD-1 signal pathways [91].

### 2.5. TIM-3 × PD-L1

T-cell immunoglobulin and mucin-domain containing-3 (TIM-3, also known as HAVCR2, CD366, KIM-3, SPTCL, and TIMD3) was originally discovered as a cell surface marker specific to IFN-γ-producing CD4^+^ T helper 1 (Th1) and CD8^+^ T cytotoxic 1 (Tc1) cells in 2002 [92]. TIM-3 is type I trans-membrane protein that contains a variable IgV domain, a glycosylated mucin domain, a single transmembrane domain, and a C-terminal cytoplasmic tail with a conserved tyrosine-based signaling motif. Interestingly, unlike other co-inhibitory molecules such as PD-1 and TIGIT, Tim-3 lacks classical inhibitory immunoreceptor tyrosine-based inhibition or immunoreceptor tyrosine-based switch signaling motifs in its cytoplasmic tail [93,94]. TIM-3 is often expressed on the surface of T cells, Tregs, macrophages, and DCs. TIM-3 expression is higher in tumor-infiltrating lymphocytes (TILs) than peripheral lymphocytes and is associated with poor patient survival [95,96]. TIM-3 can bind to at least four distinct ligands (Galectin-9, PtdSer, CEACAM1, and HMGB1) and could inhibit the Th1 response by inducing apoptosis [97,98]. The expression of TIM-3 and PD-1 was identified in highly dysfunctional CD8^+^ T cells, suggesting that co-targeting TIM-3 and PD-L1 using bsAb may enhance the antitumor benefit in tumors with a high expression of TIM-3.

Scientists from Eli Lilly and Zymeworks disclosed an IgG-like bsAb (LY-3415244) simultaneously targeting TIM-3 and PD-L1 in 2018. From November 2018 to October 2019, 12 patients were enrolled into four cohorts and received at least one dose of LY3415244. Two patients (16.7%) developed clinically significant anaphylactic infusion-related reactions and all patients developed treatment-emergent anti-drug antibodies (TE-ADAs). The ADA titers were sometimes very high and negatively impacted the soluble TIM-3 target engagement in most patients. The ADA epitope specificity was against both TIM-3 and PD-L1 arms of the bispecific antibody; most TE-ADAs initially targeted the TIM-3 arm after the first dose. Pre-existing ADAs against LY3415244 were also detected in normal (unexposed) human serum samples. One patient with PD-1 refractory non-small-cell lung cancer had a near partial response with a tumor regression of −29.6% for three consecutive scans, and this was sustained at study discontinuation. Due to unexpected immunogenicity targeting both arms of the bispecific antibody, the phase I, multicenter, open-label study of LY3415244 in patients with advanced solid tumors was terminated during dose escalation after a determination was made that the risk:benefit ratio no longer favored continued evaluation [99,100].

### 2.6. TIGIT × PD-L1

T-cell immunoreceptor with Ig and ITIM domains (TIGIT, also known as VSIG9, VSTM3, and WUCAM), identified in 2009, belongs to the poliovirus receptor (PVR)/Nectin family. Its cytoplasmic region contains an immunoglobulin tyrosine tail (ITT)-like motif and an ITIM domain responsible for TIGIT signaling [101]. TIGIT is expressed on the surface of activated CD8^+^ T cells, memory and regulatory CD4^+^ T cells, NKs, Tregs, and follicular CD4^+^ T cells, and is also co-expressed with PD-1 on tumor-antigen-specific T cells and TILs in various cancer patients [102]. TIGIT binds to at least two nectin family members, CD155 (PVR, Necl-5) with high affinity and CD112 (PVRL2, Nectin-2) with a lower affinity. Because CD155 can bind to CD226, a receptor expressed on T cells and NKs, and result in the stimulation of T cells and NKs, TIGIT competes with CD155, leading to an overall immunosuppressive phenotype and poor overall survival in cancer patients [103]. TIGIT also directly exerts T-cell-intrinsic inhibitory effects via the recruitment of the phosphatases of Src homology 2 domain-containing inositol polyphosphate 5-phosphatase 1 (SHIP1) and Src homology 2 domain-containing inositol polyphosphate 5-phosphatase 2 (SHIP2) [104]. Since TIGIT is also co-expressed with PD-1 on tumor-antigen-specific T cells and TILs in various cancer patients, co-targeting TIGIT and PD-L1 using bsAb may overcome resistance to anti-PD-L1 therapy and enhance the antitumor benefit in tumors showing a high expression of TIGIT, CD155, and CD112.

Researchers from Legend Biotech disclosed an IgG-like bsAb (BTP-21) simultaneously targeting TIGIT and PD-L1 in 2019. BTP-21 comprised humanized monoclonal IgG1κ antibody targeting PD-L1 (h53C1) fused at the N-terminus of the heavy chain to the C-terminus of two humanized sdAb targeting TIGIT via the mutated human IgG1 hinge region. BTP-21 bound to hPD-L1, human TIGIT, and mouse TIGIT with a Kd value of 5 × 10^−10^, 3 × 10^−10^, and 7.9 × 10^−10^ M, respectively. BTP-22 inhibited the binding of PD-1 to PD-L1-expressing CHO cells and the binding of CD155 to TIGIT-expressing CHO cells with an EC_50_ value of 3.37 and 3.10 nM, respectively. The incubation of PD-L1/CD155 target cells (cells expressing PD-L1 and CD155) with BTP-21 significantly blocked the binding of PD-L1 to PD-1 and TIGIT to CD155. Furthermore, the incubation of PBMC-derived primary T cells with BTP-21 resulted in the binding of the said antibody to CD8 and CD4^+^ T cells. It was also observed that BTP-22 also induced the release of IFN-γ when incubated with GS-C2/PD-L1 target cells. The administration of BTP-21 to a colon cancer MC38 mouse model resulted in decreased tumor growth [105].

Researchers from Shanghai Henlius Biotech disclosed an IgG-like bsAb (HLX-301) simultaneously targeting TIGIT and PD-L1 in 2021. HLX-301 comprised an anti-PD-L1 IgG1 antibody and two anti-TIGIT sdAbs. HLX-301 are capable of binding to TIGIT and PD-L1, bringing CD8^+^ T cells to the tumor cells, which lead to an enhanced activation of CD8^+^ T cells by the increased interaction between HLA-I on the tumor cells and TCRs on T cells. HLX-301 showed significant anti-tumor efficacy with a TGI value of 88% in hTIGIT knock-in mice with a MC38 murine colon cancer model. Two phase I/II clinical trials have been initiated to evaluate the safety, tolerability, pharmacokinetic characteristics, and efficacy of HLX301 in locally advanced/metastatic solid tumors, lymphoma, and NSCLC [106].

Biologists from Huaota Biopharm disclosed an IgG-like bsAb (HB-0036, also known as 900693) simultaneously targeting TIGIT and PD-L1 in 2022. HB-0036 comprised an anti-PD-L1 IgG1 antibody and two anti-TIGIT scFvs linked to the C-terminus of the heavy chain of the anti-PD-L1 antibody. The scFv variable kappa region and VH domains of anti-TIGIT domain were tethered to the IgG1 mAb by glycine-rich flexible (GGGGS)_4_ linkers. HB-0036 enhances the expression of the activating receptor CD226, and substantially increases T-cell proliferation and cytokine production when stimulated, compared to T cells exposed to the combination of the two parental mAbs. HB-0036 allows a greater accumulation of anti-TIGIT Ab at the tumor site through the co-engagement of PD-L1 on tumor cells with the anti-PD-L1 arm of bsAb, leading to high local drug concentrations, when compared to combination therapy with the two parental antibodies. In pre-clinical tumor models, HB-0036 substantially improves tumor control and is associated with favorable anti-tumor signatures, including the reduced infiltration of neutrophils, lower abundance of TIGIT^+^ T cells, and increased production of IFN-γ by TILs, when compared to combination therapy with the two parental antibodies [107]. Based on these results, a phase I/II clinical trial has been initiated to evaluate the safety and efficacy of HB-0036 in subjects with advanced solid tumors.

Biologists from I-Mab Biopharma disclosed an IgG-like bsAb (TIGIT-Fc-93-VH6) simultaneously targeting TIGIT and PD-L1 in 2022. TIGIT-Fc-93-VH6 consists of two anti-PD-L1 sdAbs, fused to the C-terminus of the heavy chains of the anti-TIGIT monoclonal IgG1 antibody. TIGIT-Fc-93-VH6 bound to both hPD-L1 (KD = 3.97 × 10^−9^ M) and human TIGIT (KD = 1.12 × 10^−10^ M). TIGIT-Fc-93-VH6 resulted in a potent inhibition in TIGIT/CD155 signaling, significantly enhanced T-cell activation, and the increased production of IFN-γ [108].

Biologists from Jiangsu Hengrui disclosed an IgG-like bsAb (P-O-T) simultaneously targeting TIGIT and PD-L1 in 2022. P-O-T is a bispecific IgG1 antibody with the replacement of CH1/CL with a titin-T chain and an obscurin-O chain, respectively, so as to reduce the mismatch of light and heavy chains during the manufacturing process. P-O-T bound with high affinity to hPD-L1 (KD = 9.70 × 10^−11^ and 9.54 × 10^−11^ M) and hTIGIT (KD = 2.57 × 10^−11^ and 3.03 × 10^−11^ M). The incubation of tuberculin-stimulated human PBMCs with P-O-T resulted in the increased secretion of IFN-γ. In hPD-L1- and TIGIT-expressing murine colon cancer (MC38-HL1)-cell-grafted mice, the administration of P-O-T strongly inhibited tumor growth (55%) [109].

Scientists from Nanjing Sanhome Pharmaceutical disclosed an IgG-like bsAb (HuPL721-T2353-scFab) simultaneously targeting TIGIT and PD-L1 in 2022. The first scFab (derived from humanized monoclonal HuPL7-21) targeting PD-L1, whose light chain is linked to the N-terminus of the heavy chain, further fused to IgG4 Fc with S228P, Y349C, T366S, L368A, and Y407V (knob) mutations. The second scFab (derived from humanized monoclonal HuT23-53) targeting TIGIT, whose light chain is linked to the N-terminus of the heavy chain, further fused to Fc of IgG4 with S228P, S354C, and T366W (hole) mutations. In vitro, the co-culturing of HuPL721-T2353-scFab with CHO-K1 cells expressing hPD-L1 resulted in a significant binding ability to hPD-L1 and TIGIT with an EC_50_ value of 1.77 and 1.30 nM. HuPL721-T2353-scFab exhibited a significant binding affinity to hPD-L1 and hTIGIT proteins with the KD values of 1.32 × 10^−9^ and 3.06 × 10^−9^ M, respectively. In vivo, the administration of HuPL721-T2353-scFab to a Raji-hPD-L1-lymphoma-implanted NSG mouse model resulted in better anti-tumor activity. The administration of HuPL721-T2353-scFab to MC38-hPD-L1-xenografted mouse model resulted in a TGI value of 50.7% [110].

Scientists from Biotheus Inc disclosed an IgG-like bsAb (PM-1022) simultaneously targeting TIGIT and PD-L1 in 2023. PM-1022 was engineered with a fully human monoclonal IgG1 antibody targeting TIGIT, fused at its C-terminus with a single-domain antibody (nanobody, V_HH_) targeting PD-L1. PM-1022 efficiently blocked the interaction between TIGIT and PVR/PVRRL2, and likewise PD-L1 to PD-1. PM-1022 induced higher luciferase signals than the anti-TIGIT or anti-PD-L1 mAbs alone in a luciferase reporter-based cell system and enhanced IFN-γ production in an MLR assay. In vivo, PM-1022 demonstrates similar anti-tumor efficacy to the combination of anti-TIGIT and anti-PD-L1 mAbs, which is stronger than the single agents alone. The scientists have also completed GLP toxicity studies that have shown excellent safety [111]. A phase I/II clinical trial has been initiated to evaluate the tolerance, safety, pharmacokinetic characteristics, and preliminary efficacy of PM-1022 in patients with advanced tumors in 2023.

## 3. Anti-PD-L1-Based bsAbs Targeting Co-Stimulatory Molecules

### 3.1. CD28 × PD-L1

CD28 (also known as Tp44), discovered in 1986 [112,113], is the first co-stimulatory molecule for cancer immunotherapy. CD28 is a homo-dimeric type I transmembrane protein expressed on the surface of T cells which has a single extracellular Ig-V-like domain, a transmembrane domain, and a cytoplasmic domain, that contain critical signaling motifs. CD28 expression is restricted to T cells including those that express PD-1 or CTLA-4. CD28 is the receptor for CD80 and CD86. The binding of CD28 to CD80 or CD86 provide important co-stimulatory signals for T-cell activation and expansion, cytokine production, target cell lysis, and the formation of long-lived memory [114]. Since CD28 is a co-stimulatory molecule on the process of T-cell activation while PD-L1 is a co-inhibitory molecule expressed on the surface of tumor cells, co-targeting CD28 and PD-L1 using bsAbs may enhance antitumor activity via the dual pathway mechanism, compared with monospecific CD28 or PD-L1 antibodies alone.

Researchers from Inhibrx disclosed an IgG-like bsAb (CX694) simultaneously targeting CD28 and PD-L1 in 2021. CX694 is a tetravalent, bispecific agonistic antibody consisting of two sdAbs targeting PD-L1 fused via a flexible linker to two camelid-derived sdAbs (1C9) targeting CD28, further fused to the Fc domain of human IgG1. Results from an in vitro assay suggested that the incubation of CX694 with CD3 (IL-2) Jurkat reporter cells and PD-L1-CHO-K1 cells led to significantly increased CD28 agonization on the CD3 (IL-2) Jurkat reporter cells in a PD-L1-dependent manner. The incubation of CX694 with adenocarcinoma human alveolar basal epithelial A549 cells resulted in the significantly increased activation of both CD4^+^ and CD8^+^ T cells and T-cell-mediated cytotoxicity. Moreover, CX694 treatment did not induce the robust production of TNFα by CD4^+^ T cells, indicating a reduced risk of in vivo cytokine release syndrome [115].

Scientists from Janux Therapeutics disclosed an IgG-like bsAb (Ab-1) simultaneously targeting CD28 and PD-L1 in 2022. Ab-1 is a bispecific antibody comprising Fab targeting hPD-L1, fused to a scFv targeting hCD28. Co-culture of human PBMCs with Ab-1 in the presence of target cells significantly resulted in the activation of T cells, as determined by the secretion of IFN-γ, IL-2, and TNFα. Ab-1 significantly bound to its targets PD-L1 and CD28 with KD values of 0.4 and 3.5 nM, respectively, as determined by biolayer interferometry. Furthermore, Ab-1 bound to hPD-L1 and CD28 with EC_50_ values of 0.30 and 1.27 nM, respectively. Moreover, the incubation of PBMCs with Ab-1 and the anti-TROP2 × CD3 T-cell engager in the presence of target cancer cells significantly activated the T cells and resulted in the killing of the target cells. The treatment of cynomolgus monkeys with Ab-1 demonstrated a maximum serum concentration (Cmax) of 10.73 nM, T1/2 of 2.15 h, clearance rate of 12.09 mL/hr/kg, and seven-day AUC of 843 nM × min. Moreover, the treated animals did not exhibit any signs of liver toxicity, as determined by the expression analysis of AST and ALT. In addition, the cytokine expression analysis for IFN-γ, TNFa, IL-6, IL-2, IL-4, and IL-5 was assessed in cynomolgus monkeys [116].

Researchers from Xencor disclosed an IgG-like bsAb (XENP-36764) simultaneously targeting CD28 and PD-L1 in 2022. XENP-36764 is a bispecific monoclonal IgG1 antibody (Fab-scFv-Fc format) comprising Fab targeting PD-L1 and a scFv targeting CD28, fused to a heterodimeric modified Fc domain via (GKPGS)_4_ linker. In an in vivo assay, the treatment of a cancer-induced cynomolgus monkey model with XENP-36764 significantly induced T-cell proliferation against tumor cells. XENP-36764 demonstrated a good PK profile [117].

### 3.2. CD27 × PD-L1

CD27 (also known as TNFRSF7) was identified as a TNFR family molecule on human T cells by van Lier in 1987 [118]. CD27 is a disulfide-linked 55 kDa homodimer typically existing as a glycosylated, type I transmembrane protein, frequently in the form of homodimers with a disulfide bridge linking the two monomers. The disulfide bridge is in the extracellular domain and close to the membrane [119]. CD27 is exclusively expressed on mature thymocytes including CD4^+^ and CD8^+^ peripheral blood T cells, NKs, and B cells [120]. CD27 is also highly expressed on B-cell non-Hodgkin’s lymphomas and B-cell chronic lymphocytic leukemias [121]. CD27 may also be expressed in soluble forms [122,123]. CD27 has a single ligand, CD70, which is also a TNF family member [124]. CD27 was originally defined as a human T-cell co-stimulatory molecule that increments the proliferative response to TCR stimulation [118]. The presence of CD70 dictates the timing and persistence of CD27-mediated co-stimulation. The transgenic expression of CD70 in iDCs sufficed to convert immunological tolerance to viruses or tumors into CD8^+^ T-cell responsiveness. Likewise, agonistic soluble CD70 promoted the CD8^+^ T-cell response [125] and, in CD70 transgenic mice, the CD4^+^ and CD8^+^ effector cell formation in response to TCR stimulation was greatly facilitated [126,127]. In mouse lymphoma models, tumor rejection was improved upon CD70 transgenesis or injection of CD27 antibodies [128,129].

Researchers from Celldex Therapeutics disclosed an IgG-like bsAb (CDX-527) simultaneously targeting CD27 and PD-L1 in 2019. CDX-527 is a tetravalent bispecific antibody construct comprising a fully human monoclonal IgG1κ antibody targeting PD-L1 (9H9) fused at the C-terminus of the heavy chain to the scFv of anti-CD27 monoclonal antibody 2B3. The incubation of CDX-527 with CD27 and NFκB-expressing cells established the activation of NFκB. The co-incubation of CDX-527 with PBMC-derived CD4^+^ T cells and DCs resulted in significant mixed lymphocytes responses. The incubation of PBMC-derived CD3^+^ T cells with CDX-527 resulted in the increased production of IL-2 by the T cells. The administration of CDX-527 to transgenic hCD27-expressing (HuCD27-Tg) mice with lymphoma (BCL1) cells resulted in a significant increase in the number of T cells (CD8^+^ and CD4^+^) and expression of IFN-γ and granzyme B. In an in vivo pharmacokinetics assay, the administration of CDX-527 to non-human primates (NHPs) resulted in an increased plasma concentration and half-life [130].

### 3.3. OX40 × PD-L1

OX40 (also known as CD134 and TNFRSF4) was discovered by Paterson in 1987 [131]. OX40 is a co-stimulatory molecule that belongs to the TNF receptor superfamily. OX40 contains an extracellular domain with a typical signal sequence, a single putative transmembrane sequence with 25 predominantly hydrophobic amino acids, and a cytoplasmic domain with 36 amino acids. The sequence of the extracellular domain includes a cysteine-rich region [132]. OX40 is expressed on diverse T-cell subsets, NKs, natural killer T cells (NKTs), and neutrophils, while its ligand, OX40L (CD252), is expressed on activated APCs such as DCs, B cells, and macrophages. Upon activation by TCR-MHC-peptide interaction, a OX40L homotrimer is formed and binds to three OX40 receptors to result in receptor crosslinking. The clustered OX40 recruit tumor necrosis factor receptor-associated factors (TRAFs) to the intracellular domain of OX40 [133]. TRAF 2 and 3 activate the PI3K/PKB, NFκB1, and NFAT pathways that account for T-cell activation, survival, and cytokine production [134]. Thus, OX40 has the potential to augment proliferation, suppress apoptosis, and induce a greater cytokine response from T cells. As T cells must first be activated to express OX40, monotherapy using an OX40 agonist may not be the best setting for testing immunotherapeutic agents [135]. BsAbs targeting OX40 and PD-L1 may be an alternative therapeutic approach, with enhanced antitumor activity, compared with monospecific OX40 or PD-L1 antibodies.

Biologists from Inhibrx disclosed an IgG-like bsAb (28A10 × 2E4) simultaneously targeting OX40 and PD-L1 in 2017. The 28A10 × 2E4 is a bispecific construct consisting of two camelid sdAb fused to an IgG-Fc domain targeting OX40 and PD-L1, respectively. Transfection of PD-L1-positive CHO cells with 28A10 × 2E4 resulted in the decreased NFκ-light-chain-enhancer of activated B-cell activities (NFκB, tumor-mediator), indicating its anti-tumor activity [136].

Scientists from Innovent Biologics disclosed an IgG-like bsAb (IBI-327, also known as anti-PD-L1/OX40 bsAb) simultaneously targeting OX40 and PD-L1 in 2020. IBI-327 is a bispecific antibody comprising two polypeptides, wherein each polypeptide consists of anti-human OX40 IgG2 monoclonal antibody fused at its C-terminus to anti-PD-L1 humanized single-domain antibody (nanobody) via a (GGGGS)_2_ flexible linker. IBI-327 bound to human OX40 and hPD-L1 with KD values of 9.98 × 10^−8^ M and 3.71 × 10^−9^ M, respectively. IBI-327 bound to CHO cells (overexpressing human OX40 and hPD-L1) with EC_50_ values of 8.46 nM and 8.73 nM, respectively. IBI-327 bound to human T cells with an EC_50_ value of 4.06 nM. Co-culture of IBI-327 with mature DC or Raji PD-L1 cells and prestimulated CD4^+^ T cells resulted in the increased production of IL-2 and IFN-γ. Human lung cancer cells (NCI-H292) treated with IBI-327 had activated the NFκB signaling pathway. Human colon-cancer-cell-induced NPG mice treated with IBI-327 had induced a tumor inhibitory effect with a TGI value of 80%. Furthermore, NCI-H292-induced NOG mice treated with IBI-327 had induced a tumor inhibitory effect with TGI values of 62–94%. Moreover, IBI-327 increased the infiltration of CD3 and Treg cells in the tumor of mice. Taken together, these findings demonstrate that IBI-327 induced enhanced immune activation and improved cancer immunotherapy [137].

Biologists from AstraZeneca and MedImmune disclosed an IgG-like bsAb (MEDI-1109) simultaneously targeting OX40 and PD-L1 in 2020. MEDI-1109 is composed of an OX40 agonist molecularly fused to two scFvs derived from a PD-L1 antagonist. MEDI-1109 retained the functional properties of each parent Ab, which included FcγR clustering of OX40 to enhance T-cell co-stimulation, binding, and full blockage of the PD-L1/PD-1 pathway, overcoming regulatory T-cell suppression, as well as the ADCC-mediated depletion of intratumoral OX40^+^ Tregs. In addition to these properties, the new bispecific antibody presented novel mechanisms of action. These included OX40 agonism mediated by PD-L1 on tumor or immune cells to enhance the T-cell co-stimulation, directed intratumoral targeting, and enhanced intratumoral T-cell co-stimulation, and, finally, ADCC-mediated depletion of PD-L1^+^ tumor cells. In vitro, MEDI-1109 demonstrated greater activity than the anti-OX40 and anti-PD-L1 combination. When evaluated in non-GLP cynomolgus monkeys, it was observed that treatment with MEDI-1109 resulted in reduced plasma sPD-L1 levels. The mutation of Fc improved the pharmacokinetics of the bispecific antibody, resulting in a two-fold reduction in clearance compared to the bispecific antibody with wild-type Fc (Cl = 12.4 versus 27.1 mL/kg/day, respectively). The mutated Fc also resulted in a two-fold increase in half-life and AUC. These findings suggest that the novel bivalent bispecific antibody could overcome the limitations of OX40 agonists in a clinical setting [138].

Scientists from JN Biosciences disclosed an IgG-like bsAb (BS-813) simultaneously targeting OX40 and PD-L1 in 2020. BS-813 is a bispecific antibody targeting hPD-L1 and hOX40. The incubation of BS-813 with phytohemagglutinin-L (PHA-L) and Staphylococcus enterotoxin B (SEB)-treated PBMCs resulted in increased IL-2 [139].

Researchers from Jiangsu Alphamab disclosed an IgG-like bsAb (KN-052) simultaneously targeting OX40 and PD-L1 in 2021. KN-052 comprises a PD-L1 binding moiety and an OX40 binding moiety, wherein the OX40 binding moiety is capable of binding amino acid residues G70 and/or F71 in a human OX40 extracellular domain and the PD-L1 binding moiety is capable of binding amino acid residues I54, Y56, E58, Q66, and/or R113 in an N-terminal IgV domain of hPD-L1. KN-052 significantly bound to A375-hPD-L1 and HEK293-OX40 cells and PBMC-derived activated CD4^+^ T and CD8^+^ T cells. Moreover, KN-052 significantly blocked the interaction between PD-L1/PD-1 and PD-L1/CD80. In addition, KN-052 effectively relieves the inhibitory effect caused by PD-L1/PD-1 binding in the NFAT system of hPD-L1-expressing CHO cells. The incubation of KN-052 with PD-L1-expressing HEK293 and OX40-expressing HEK293 cells resulted in an increased ADCC of the target cells. ELISA assays indicated the incubation of KN-052 with human PBMCs resulted in an enhanced immune response, as observed by the increased IL-2 levels. The administration of KN-052 to a mouse colon cancer hPD-L1-MC38-cell-injected mouse model resulted in increased tumor growth inhibition. In ae pharmacokinetic assay, the administration of KN-052 to cynomolgus monkeys resulted in ae half-life between 69 to 99 h and Cmax between 26,045 to 318,177 mcg/L [140].

Researchers from EpimAb Biotherapeutics disclosed an IgG-like bsAb (EMB-09, also known as FIT1014-20a) simultaneously targeting OX40 and PD-L1 in 2022. EMB-09 is a bispecific Fabs-In-Tandem Immunoglobulin (FIT-Ig) consisting of human OX40-targeting arm and hPD-L1-targeting arm. EMB-09 bound to PD-L1 and OX40 (KD = 2.55 × 10^−10^ and 6.02 × 10^−9^ M, respectively). Similarly, EMB-09 bound to hPD-L1-expressing CHO cells and OX40-expressing CHO cells (EC_50_ = 1.8 nM). In addition, EMB-09 cross-reacted with cynomolgus monkey PD-L1 and OX40 (KD = 5.87 × 10^−10^ and 5.13 × 10^−9^ M, respectively). EMB-09 blocked PD-1 protein binding to PD-L1-over-expressing CHO cells. The incubation of the T-cell activating molecule and PD-L1-expressing CHO cells (CHO-PD-L1-OS8) and human PD-1- and NFAT-expressing Jurkat cells (Jurkat-PD-1-NFAT-luciferase reporter cells) with EMB-09 resulted in the inhibition of PD-L1-mediated PD-1 downstream signaling, as determined by an NE-Glo luminescence assay. The incubation of PD-L1-expressing CHO cells with Jurkat-OX40-NFκB-luciferase reporter cells with EMB-09 resulted in the dose-dependent activation of the OX40 downstream NFκ signaling pathway, indicating that the EMB-09-induced activation is PD-L1-dependent. Co-culturing of CHO-PD-L1-OS8 cells and human PBMC-derived primary T cells with EMB-09 resulted in the activation of T cells, indicated by the increased secretion of both IFN-γ and IL-2. The incubation of PBMC-derived monocyte, mature DCs, and allogeneic human CD4^+^ T cells with EMB-09 resulted in a potent secretion of T-cell-activating IL-2. Similarly, enhanced T-cell activation was observed on the incubation of Staphylococcus-aureus-enterotoxin-B-stimulated T cells with EMB-09. Further, EMB-09 induced no CDC against human CD4^+^ T cells, and induced little phagocytosis effect against CD14^+^ monocytes, contributing to the safety of the drug. In colon cancer cell (MC38)-grafted hPD-L1/hOX40-expressing mice, and colon cancer cell (CT26) human PD-1/PD-L1/OX40-expressing mice, the administration of EMB-09 strongly inhibited tumor growth, as compared to atezolizumab [141].

Researchers from Anhui Anke Biotechnology disclosed an IgG-like bsAb (L52-2D7H232) simultaneously targeting OX40 and PD-L1 in 2023. L52-2D7H232 is a fully human bispecific antibody comprising a monoclonal IgG1 domain targeting PD-L1 and a variable region nanobody targeting OX40. In vitro, L52-2D7H232 exhibited a significant binding affinity towards PD-L1 and OX40, and blocked the binding of OX40L to the OX40 and PD-1/PD-L1 signaling pathway. The agonist activity of L52-2D7H232 was assessed by the promotion of NFκB-mediated transcriptional activation in a luciferase reporter T-cell activation assay. L52-2D7H232 exhibited significant PD-L1 dependent activity in the DG44-PDL1-FL-4B6 and Jurkat-OX40-NFκB-21C6 assay systems. Furthermore, L52-2D7H232 had strong binding activity for cynomolgus monkey OX40 and human OX40, and no binding affinity for mPD-L1 was observed by an ELISA assay. The administration of L52-2D7H232 to humanized mice (C57) xenografted with mouse colon adenocarcinoma cells (MC38-PD-L1) resulted in a significant inhibition of tumor growth, and the complete elimination of the tumor showed their potent anti-tumor activity in mice [142].

### 3.4. CD137 × PD-L1

CD137 (also known as 4-1BB and TNFRSF9), a co-stimulatory molecule that belongs to the TNF receptor superfamily, was cloned from activated T cells in 1989 [143]. Besides activated T cells, CD137 is expressed on multiple lineages of hematopoietic cells, including regulatory T cells, B cells, NKs, monocytes, and DCs [144]. CD137 is composed of 255-amino acids having a short N- terminal cytoplasmic portion, a transmembrane region, and an extracellular domain that contains three cysteine-rich motifs [145]. The ligation of CD137 by its ligand CD137L (also known as 4-1BBL and TNFSF9), which is mainly, though not exclusively, expressed on antigen-presenting cells (APCs), evokes various T-cell responses such as cell expansion, increased cytokine secretion, and the prevention of activation-induced cell death. Thus, such ligation serves to activate the immune system [146]. The art has recognized the benefit of providing a combination immunotherapy involving binding molecules specific for CD137 and binding molecules specific for PD-L1. BsAbs targeting CD137 and PD-L1 can induce the survival and proliferation of T cells, thereby enhancing the anti-tumor immune response.

Researchers from Inhibrx and Elpiscience BioPharma disclosed an IgG-like bsAb (ES-101, also known as INBRX-105-1) simultaneously targeting CD137 and PD-L1 in 2017. ES-101 consisted of two sdAbs targeting PD-L1 and two sdAbs targeting CD137, fused to the Fc domain. ES-101 displayed high binding affinities to CD137 and PD-L1 expressing 293 freestyle cells and K562 leukemia cells in vitro. This bi-specific fusion protein induced the proliferation and activation of CD8^+^ T cells and IFN-γ secretion in a culture comprising immature DCs and enriched T cells. ES-101 also exhibited far superior T-cell co-stimulation. ES-101 significantly bound to CD137 and PD-L1 expressing 293 freestyle cells and K562 leukemia cells. ES-101 induced high T-cell co-stimulation when incubated with T cells and autologous immature DCs [147]. However, two phase I clinical trials to evaluate the safety antitumor efficacy of ES101 in patients with advanced solid tumors have been terminated or withdrawn in 2022, due to the sponsor’s clinical development strategy adjustment.

Biologists from Merus and Incyte disclosed an IgG-like bsAb (MCLA-145, also known as PB-17311) simultaneously targeting CD137 and PD-L1 in 2018. MCLA-145 is a bispecific human IgG1 antibody targeting PD-L1 and CD137. MCLA-145 bound to PD-L1 and hCD137 with high affinity and exhibited a higher affinity for activated T cells. MCLA-145 has the ability to enhance IL-2 cytokine release by PBMCs in the presence of Staphylococcal enterotoxin B. MCLA-145 significantly increased the production of IFN-γ in M2 macrophage/PBMCs co-cultures. MCLA-145 significantly enhanced the expansion and differentiation of antigen-specific CD8^+^ T cells. MCLA-145 enhanced the proliferation of CD4^+^ and CD8^+^ TILs derived from patients with hepatocellular carcinoma and leptomeningeal metastasis colorectal cancer. Moreover, MCLA-145 bispecific antibody provided a statistically significant survival benefit for human RKO colon carcinoma in NSG mice engrafted with human PBMCs [148]. MCLA-145 is in phase I clinical development at Merus for the treatment of advanced or metastatic solid tumors. In 2017, Merus and Incyte entered into a collaboration and license agreement by which Incyte received ex-U.S. rights. In 2022, this agreement was terminated and Merus regained full global rights to the product.

Researchers from Antengene and OriCell Therapeutics disclosed an IgG-like bsAb (ATG-101, also known as YN-051) simultaneously targeting CD137 and PD-L1 in 2019. ATG-101 is a bispecific human monoclonal IgG1λ antibody targeting hCD137 and hPD-L1. In an in vitro assay, ATG-101 was observed to bind to hPD-L1 and hCD137 with KD values of 5.17 × 10^−10^ and 1.64 × 10^−9^ M, respectively. ATG-101 was found to bind to hPD-L1-expressing CHO cells substantially. ATG-101 inhibited the binding of hPD-L1/hPD-1, as analyzed by flow cytometry. ATG-101 also bound substantially well to mouse PD-L1-expressing CHO cells. ATG-101 was observed to have significant agonistic activity when incubated with human embryonic kidney 293T cells expressing hCD137. In an in vivo assay, the administration of ATG-101 to colorectal-cancer-MC38-injected C57BL/6 female mice transfected with the hCD137 gene resulted in complete tumor regression in 5 of the 6 treated mice [149]. ATG-101 is in phase I development at Antengene Biologics Limited for the treatment of metastatic/advanced solid tumors.

Biologists from ABL Bio and I-Mab Biopharma disclosed an IgG-like bsAb (ABL-503, also known as TJ-L14B) simultaneously targeting CD137 and PD-L1 in 2020. ABL-503 significantly bound to PD-L1 and CD137. The incubation of ABL-503 with PD-L1-expressing cancer cells resulted in significant CD137 activation. The co-incubation of ABL-503 with human PBMCs with mammary gland cancer cells (HCC1954) resulted in the dose-dependent activation of T cells. In an in vivo assay, the administration of ABL-503 to hPD-L1-expressing murine colorectal cancer (MC38)-cell-injected humanized mice resulted in significant tumor inhibition in these mice [150]. ABL-503 is in phase I development at ABL Bio and I-Mab Biopharma for the treatment of locally advanced or metastatic solid tumors.

Scientists from F-star Therapeutics disclosed an IgG-like bsAb (FS-222, also known as FS22-172-003-AA/E12v2) simultaneously targeting CD137 and PD-L1 in 2020. FS-222 is a bispecific IgG1 antibody comprising an Fc domain with antigen binding activity targeting hCD137 (FS22-172-003-AA), wherein part of the CH3 domain is substituted with human monoclonal IgG1κ antibody (G1-AA/E12v2) targeting hPD-L1, harboring L234A and L235A (LALA) mutations in the CH2 domain of heavy chain. FS-222 bound to hPD-L1 (KD = 0.19 nM) and cynomolgus PD-L1 (KD = 0.36 nM), dimeric hCD137 (KD = 0.7 nM), and cynomolgus CD137 (KD = 0.8 nM). Co-culturing of HEKhPD-L1 and CD8^+^ T cells with FS22-172-003-AA/E12v2 resulted in enhanced T-cell activation, as indicated by increased levels of IL-2. FS-222 exhibited an increased potency when incubated with expanded human CD4^+^ T cells and monocyte-derived dendritic cells, as indicated by the enhanced IFN-γ [151]. FS-222 is currently in phase I clinical development at F-star Therapeutics for the treatment of advanced malignancies.

Scientists from Pieris Pharmaceuticals and Servier disclosed an IgG-like bsAb (ONC-0055, also known as S-095012) simultaneously targeting CD137 and PD-L1 in 2020. ONC-0055 is a codon-optimized Fc fusion protein comprising a human monoclonal IgG4 (S228P) antibody targeting hPD-L1, fused at each HC C-terminus to a hCD137-specific lipocalin mutein. ONC-0055 was observed to bind to hPD-L1 and hCD137 with KD values of 0.59 and 6.48 nM, respectively. ONC-0055 was observed to bind simultaneously to PD-L1 and CD137 with EC_50_ values of 0.58 nM (PD-L1 capture-CD137) and 2.9 nM (CD137 capture-PD-L1). The incubation of human PBMCs with ONC-0055 resulted in significantly increased IL-2 production. In a mixed lymphocyte reaction assay, the incubation of monocyte-derived dendritic cells (moDCs) and CD4^+^ T cells with ONC-0055 resulted in the activation of CD8^+^ T cells and a dose-dependent secretion of IL-2, perforin, granzyme B, and granzyme A. ONC-0055 was observed to have a plasma terminal half-life of 295 h when administered to male CD-1 mice [152]. ONC-0055 is being co-developed by Pieris Pharmaceuticals and Servier as a treatment for advanced/metastatic solid tumors. It is undergoing phase I/II clinical evaluation for this indication in patients progressing on standard-of-care therapy.

Biologists from AP Biosciences disclosed an IgG-like bsAb (AP-203) simultaneously targeting CD137 and PD-L1 in 2020. AP-203 is a bispecific antibody comprising anti-PD-L1 monoclonal IgG antibody fused at its Fc C-terminal end to anti-CD137 scFv. The incubation of human peripheral blood-derived monocytes and allogeneic CD4^+^ T cells with AP-203 resulted in the increased activation of CD4^+^ T cells, indicated by an increase in the release of IL-2 and IFN-γ, as evaluated by the mixed lymphocyte reaction. Similarly, AP-203 enhanced T-cell activation, when co-cultured with human memory T cells (CD4^+^ and CD8^+^) and MHC-II-stimulated autologous immature cells, and co-cultured with human T cells and PD-L1-overexpressing HEK293 cells. In addition, the incubation of CD8^+^ T cells and PD-L1-positive tumor cells (NCI-H1975, PC-3, and MDA-MD-231) with AP-203 resulted in a potent lysis of cancer cells, indicated by the production of IFN-γ. Moreover, the incubation of peripheral blood-derived Treg cells with AP-203 resulted in Treg-cell-mediated inhibition in the suppression of the proliferation and IL-2 production of CD4^+^ T cells. In human tumor cells (lung cancer cells: NCI-H292 and NCI-H1975; and pancreatic cancer cells: BxPC-3) and human PBMC-xenografted mouse models, the administration of AP-203 inhibited tumor growth (TGI: 67.5%, 80%, and 43%, respectively). AP-203 exhibited a good half-life (49–87 h), and induced no toxic effect in rhesus monkeys [153,154]. AP-203 is in phase I/II development at AP Biosciences to investigate the safety, pharmacokinetics, and clinical activity of AP203 in patients with locally advanced or metastatic solid tumors, with an expansion to selected malignancies.

Biologists from Roche disclosed an IgG-like bsAb [4-1BB (20H4.9) × PD-L1] simultaneously targeting CD137 and PD-L1 in 2020. The 4-1BB (20H4.9) × PD-L1 is an agonistic bispecific IgG1 antibody comprising bivalent anti-hCD137 Fab (derived from 20H4.9), fused to a knob/hole Fc domain (harboring mutations P329G, L234A, L235A, S354C, T366W, Y349C, T366S, L368A, and Y407V) further fused via (GGGGS)_4_ linker to monovalent anti-hPD-L1 scFv. 4-1BB (20H4.9) × PD-L1 bound to gastric cancer cells (MKN45) with an EC_50_ value of 1.95 nM [155].

Researchers from BioNTech and Genmab disclosed an IgG-like bsAb (BNT-311, also known as GEN-1046) simultaneously targeting CD137 and PD-L1 in 2021. BNT-311 is a bispecific humanized monoclonal IgG1 antibody targeting PD-L1 and CD137, harboring L234F, L235E, D265A, and F405L (FEAL), and L234F, L235E, D265A, and K409R (FEAR) mutations in the Fc domain. BNT-311 simultaneously bound to hPD-L1 and hCD137 which are separately expressed on both human myelogenous leukemia (K-562) cells and CD8^+^ T cells. BNT-311 can enhance CD8^+^ T-cell proliferation in a dose-dependent manner when co-cultured with PD-L1 endogenously expressing monocytes derived from dendritic cells or tumor cells and CD137+ T cells. BNT-311 significantly induced the expansion of tumor-infiltrating lymphocytes when co-cultured with human non-small-cell lung carcinoma. In a pharmacodynamic assay, BNT-311 significantly increased serum circulating levels of IFN-γ and IFN-γ-IP-10 and enhanced effector memory CD8^+^ T-cell proliferation (by two-fold) in the peripheral blood of patients with an advanced solid tumor [156]. In a dose-escalation phase I/IIa study, patients with NSCLC were observed to have an improved complete response and confirmed disease control by 88.9% when treated with BNT-311.

Biologists from Nanjing Leads Biolabs disclosed an IgG-like bsAb (LBL-024, also known as P4B-3) simultaneously targeting CD137 and PD-L1 in 2021. LBL-024 is a tetravalent bispecific monoclonal antibody comprising two scFvs targeting hCD137 fused to the C-terminus constant region of anti-PD-L1 monoclonal IgGκ antibody. In an in vitro assay, LBL-024 bound to hPD-L1 and hCD137 with KD values of 2.89 × 10^−10^ and 1.46 × 10^−10^ M, respectively, as determined by Octet technology. In addition, LBL-024 inhibited the hPD-L1 and hCD137 with EC_50_ values of 0.03 and 0.07 nM, respectively. The incubation of LBL-024 with human allogeneic PBMCs and PD-L1-expressing human B lymphoblastoid Raji cells resulted in the significantly increased stimulation of PBMCs to release high levels of IL-2 in a dose-dependent manner. The administration of LBL-024 to hPD-L1-expressing murine colon adenocarcinoma MC38-cell-implanted Balb/c-hPD-L1/CD137 double knock-in mice resulted in a significant tumor inhibitory effect and low tumor weight with a TGI value of 81.8%. Moreover, LBL-024 elevated the CD8^+^ T-cell ratio and reduced the Treg ratio in the TME in said mice. In a pharmacokinetics (PK) study, the administration of LBL-024 to male SD rats resulted in a Cmax of 194.73 and 136.25 mcg/mL (for PD-L1 and CD137, respectively), and an AUC of 609.82 and 455.78 day × mcg/mL (for PD-L1 and CD137, respectively) [157]. LBL-024 is currently in phase I/II clinical development at Nanjing Leads Biolabs for the treatment of advanced malignancies.

Scientists from Jiangsu Hengrui disclosed an IgG-like bsAb (9EN-FM) simultaneously targeting CD137 and PD-L1 in 2021. The 9EN-FM is a recombinant, bispecific antibody comprising two scFvs targeting hPD-L1, fused to the N-terminus of a monoclonal IgG1κ antibody targeting hCD137, harboring L234A, L235A, and P329G mutations in the HC constant region. The 9EN-FM bound to hCD137 and hPD-L1 with KD values of 3.2 and 0.07 nM, respectively. The incubation of 9EN-FM with HEK293 cells expressing PD-1 and the NFAT luciferase reporter gene resulted in the dose-dependent inhibition of NFAT fluorescence signaling (mediated by the PD-1/PD-L1 signaling pathway). The incubation of 9EN-FM with PD-L1-expressing HEK293 cells and human PBMCs resulted in significantly increased PD-L1-dependent T-cell activation. The administration of 9EN-FM to PD-L1-expressing, colon adenocarcinoma (MC38-hPD-L1)-xenografted Balb/c-hPD-L1/hCD137 double-deleted mice resulted in significantly increased anti-tumor activity in a dose-dependent manner. In a pharmacokinetic study, 9EN-FM demonstrated a half-life of 37.3 h when administered to male cynomolgus monkeys [158].

Biologists from Biotheus disclosed an IgG-like bsAb (PM-1003) simultaneously targeting CD137 and PD-L1 in 2022. PM-1003 is a bispecific fusion antibody consisting of an alpaca-derived humanized sdAb targeting cysteine-rich domain 4 (CRD4) of hCD137, fused to the C terminus of human IgG1 Fc region via a 20-residue-long (GGGGS)_4_ linker, and alpaca-derived humanized sdAb targeting hPD-L1, wherein the CH2 domain of Fc region harbor a LALA (L234A/L235A) mutation expressed in HEK293F cells. PM1003 did not show any non-specific binding to non-transfected CHO cells. PM-1003 is more potent than the combination of anti-PD-L1 and anti-CD137 antibodies. T-cell activation mediated by PM1003 was dependent on crosslinking via PD-L1-mediated cross-bridging. PM-1003 substantially inhibited tumor growth in the MC38-hPD-L1 and CT-26-hPD-L1 tumor models. PM-1003 showed limited anti-CD137-mediated hepatotoxicity compared with Urelumab [159,160].

Researchers from QLSF Biotherapeutics disclosed an IgG-like bsAb (QL-301) simultaneously targeting CD137 and PD-L1 in 2022. QL-301 effectively bound to human/cynomolgus PD-L1-expressing human embryonic kidney (HEK293) cells with EC_50_ values of 1.60 and 0.73 nM, respectively, as determined by flow cytometry. In an NFκB reporter assay, the incubation of PD-L1-expressing HEK293 cells with QL-301 in the presence of CD137 resulted in the effective activation of CD137. The incubation of PD-L1-expressing epidermoid cancer (A431) cells and human PBMCs with QL-301 resulted in the enhanced production of IL-2 and IFN-γ and induced CD8^+^ T-cell proliferation (in the presence of anti-CD3 antibody). In an in vivo assay, the treatment of PD-L1-expressing colon cancer (MC38)-cell-engrafted C57BL/6 mice model with QL-301 resulted in increased CD8^+^ T-cell numbers and inhibited tumor growth. In a toxicology study, QL-301 demonstrated fewer toxic effects on liver tissues, as determined by asparate transaminase and alanine transaminase levels, when administered to rhesus monkey [161].

Researchers from Merus disclosed an IgG-like bsAb (MF6797 × MF7702) simultaneously targeting CD137 and PD-L1 in 2022. MF6797 × MF7702 is a bispecific IgGκ antibody comprising common light chains, a VH targeting the ECD of hPD-L1, and a second VH targeting hCD137 ECD, wherein the Fc domain harbors a L351K and T366K (KK)/L351D and L368E (DE) CH3 heterodimerization domain. The incubation of fresh tumor explants containing tumor-specific Teffs and Tregs with MF6797 × MF7702 resulted in increased interferon IFN-γ production in the tumor tissues. The administration of MF6797 × MF7702 in lung- and breast-cancer-implanted NSG mice resulted in increased tumor growth inhibition, which was associated with a skewed distribution of Ly95 T cells and increased NY-ESO antigen-specific T cells within the tumor tissues [162].

Scientists from Hanmi Pharmaceutical disclosed an IgG-like bsAb (BH-3120) simultaneously targeting CD137 and PD-L1 in 2022. BH-3120 is a heterodimeric bispecific IgG antibody comprising two dimers of light and heavy chain linked via a disulfide bond targeting PD-L1 and CD137, respectively. BH-3120 blocked the binding of PD-L1 and PD-1 and PD-L1 to CD80. In addition, the treatment of human T cells with BH-3120 under the stimulation of allogeneic dendritic cells resulted in strong T-cell regulatory activity and promoted the secretion of IL-2. BH-3120 further crosslinked PD-L1 on human colon cancer (DLD-1) cells expressing PD-L1 and CD137 on GS-H2 cells expressing CD137, producing IL-8, thereby mediating CD137 receptor agonistic activity. In vivo, the administration of BH-3120 to hPD-L1/hCD137 double-derived mice xenografted with mice colon cancer (MC38) cells resulted in strong antitumor efficacy. In hPD-1/hCD137 double-derived mice xenografted with MC38 cells, BH-3120 and anti-PD-1 antibody (BH-2917b) significantly inhibited tumor growth with a TGI value of 122%. In another assay, the administration of BH-3120 and BH-2917b to mouse (CT-26)-xenografted hPD-1/hPD-L1/hCD137 triadic mice resulted in a TGI value of 79.6%, thereby treating cancer [163].

Biologists from Eutilex disclosed an IgG-like bsAb (EuPD-1 × 94 kvt LHC218) simultaneously targeting CD137 and PD-L1 in 2022. EuPD-1 × 94 kvt LHC 218 is bispecific antibody consisting of two euPD-1 peptides targeting hPD-L1, fused via (GGGGS)_2_ linker to IgG1 Fc that further fused two 94 kvt-derived scFvs targeting CD137. A surface plasmon resonance assay indicated euPD-1 × 94 kvt LHC218 bound to PD-L1 antigen with a KD value of 1.58 nM. Moreover, the co-incubation of MDA-MB-231 with PD1- and CD137-positive effector cells with euPD-1 × 94 kvt LHC218 dose-dependently activated CD137 and inhibited the PD-1/PD-L1 interaction [164].

Researchers from Harbour BioMed disclosed an IgG-like bsAb (PR-004270) simultaneously targeting CD137 and PD-L1 in 2022. PR-004270 is a bispecific monoclonal IgG1λ antibody comprising Fab targeting PD-L1 and heavy chain variable domain antibody targeting CD137, harboring a C220S mutation in the CH2 region and L234A and L235A mutations in the Fc domain. PR-004270 bound to hPD-L1-expressed CHO-K1 cells with an EC_50_ value of 0.96 nM, as determined by flow cytometry and activated T cells releasing IFN-γ. Similarly, PR-004270 bound to hCD137-expressing CHO-K1 cells with an EC_50_ value of 1.58 nM. Furthermore, PR-004270 showed strong T-cell activation activity and an increased release of IFN-γ and IL-2. PR-004270 showed a half-life of 465.6 h, apparent volume of distribution of 75.7 mL/kg, AUC of 17,536 h × mcg/mL, and clearance (Cl) of 0.11 mL/h/kg, when administered to BALB/c female mice [165].

Researchers from Anhui Anke Biotechnology disclosed an IgG-like bsAb (HK-010) simultaneously targeting CD137 and PD-L1 in 2023. HK-010 is an Fc-silenced humanized IgG4 bispecific antibody comprising a first Fab targeting PD-L1, linked to an IgG4 Fc domain that is further linked to a scFv targeting CD137 via a linker. HK-010 significantly bound to hPD-L1 and hCD137 with EC_50_ values of 0.11 and 0.21 nM, respectively. HK-010 bound to hPD-L1 and hCD137 expressed on CHO-K1 cells with EC_50_ values of 7.60 and 16.39 nM, respectively. The incubation of human PBMCs with HK-010 significantly activated T cells and increased the secretion of IL-2. The incubation of human CD8^+^ T cells and CHO-K1/hPD-L1 cells in the presence of HK-010 significantly activated the T cells and increased the secretion of IFN-γ. The incubation of HEK-293/NFκB-Luci/CD137 cells and CHO-K1/hPD-L1 cells with HK-010 (20 mcg/mL) significantly activated the NFκB signaling pathway downstream of CD137 depending on the concentration of PD-L1. The treatment of MC38-implanted, PD-1/CD137 double-humanized C57BL/6 mice with HK-010 demonstrated anti-tumor efficacy (TGI, 99.87% for 2 mg/kg). The administration of HK-010 to rhesus monkeys demonstrated the safety of the product, wherein alanine aminotransferase and aspartate transaminase was not significantly increased and white blood cells (WBCs) and lymphocytes were also found to be in the normal range [166].

Researchers from Novarock Biotherapeutics disclosed an IgG-like bsAb (1923Ab18) simultaneously targeting CD137 and PD-L1 in 2023. The 1923Ab18 is a bispecific antibody comprising a heavy chain containing an anti-PD-L1 antibody-derived heavy chain, fused to an anti-CD137 scFv, and the light chain consisting of an anti-PD-L1 antibody light chain. The 1923Ab18 co-incubated with PD-L1^+^ HEK293T cells and Jurkat cells induced PD-L1-dependent CD137 signaling in these cells. Moreover, 1923Ab18 significantly blocked the interaction between PD-1 and PD-L1. In addition, 1923Ab18 co-incubated with CD8^+^ T cells and PD-L1^+^ NUGC-4 gastric tumor cells induced T-cell activation and cytotoxicity against the treated tumor cells. The 1923Ab18 bound to PD-L1 and CD137 with high affinities (KD = 3.49 × 10^−9^, and 2.23 × 10^−8^ M), as determined by a Biacore assay. An in vivo assay suggested that the administration of 1923Ab18 to a PD-L1^+^ MC38 colon-cancer-cell-induced tumor mouse model resulted in the inhibition of tumor growth in these mice. The treatment further increased the infiltration of CD8^+^ cells and reduced the level of Tregs in the TME of these mice [167].

### 3.5. ICOS × PD-L1

Inducible T-cell co-stimulator (ICOS, also known as CD278 and CVID1) was identified by Hutloff A in 1999 [168]. It is a 55 kDa transmembrane protein that consists of a single IgV-like domain, a transmembrane region, and a cytoplasmic tail. The ICOS amino-acid sequence shares 24% (17%) identity and 39% (39%) similarity with CD28 (and CTLA-4). The cysteine residue located at position 141 of CD28, also found in CTLA-4, is apparently involved in forming the disulfide bridge between the homodimeric chains of these proteins, and is also found in ICOS (position 136). ICOS is exclusively expressed on T lymphocytes, and is found on a variety of T-cell subsets, while its ligand (ICOSL, also known as B7-H2) is expressed on B cells and APCs [169]. It is present at low levels on naive T lymphocytes but its expression is rapidly induced upon immune activation and upregulated in response to pro-inflammatory stimuli such as on engagement of TCR and co-stimulation with CD28 [170,171]. ICOS plays an important role in the late phase of T-cell activation, memory T-cell formation, and the regulation of humoral responses through T-cell-dependent B-cell responses [172,173]. As a co-stimulatory molecule, it serves to regulate TCR-mediated immune responses and antibody responses to antigen.

Scientists from Kymab disclosed an IgG-like bsAb (KY-1055, also known as STIM-003_289) simultaneously targeting ICOS and PD-L1 in 2017. KY-1055 is a tetravalent bispecific human monoclonal IgG1 antibody targeting hICOS and hPD-L1. The binding affinity of KY-1055 towards hPD-L1 and hICOS was found to be 0.26 and 0.70 nM, respectively. The EC_50_ value of KY-1055 for hICOS and hPD-L1 was 0.63 and 1.43 nM, respectively. In addition, KY-1055 bound to CHO cells expressing hICOS and hPD-L1 with EC_50_ of 4.92 × 10^−9^ M and 0.64 nM, respectively. KY-1055 induced ADCC against CHO cells expressing hICOS in the presence of Jurkat effector cells (EC_50_ = 2.80 × 10^−10^ M). It also induced ADCC against ICOS-expressing CCRF-CEM cells by PBMC-derived NK cells (EC_50_ = 12 pM). KY-1055 neutralized the binding of hICOS and hPD-L1 to their respective targets with an IC_50_ of 0.81 and 0.28 nM. KY-1055 activated the T cells, as indicated by the increase in the production of IFN-γ (EC_50_ = 8.10 nM) [174].

## 4. Conclusions

Monospecific anti-PD-L1 antibodies have shown durable clinical benefits and long-term remissions in patients with various cancers. However, this kind of antibody also has some disadvantages including a low response rate in patients (around 20%). Combination therapies of anti-PD-L1 antibodies with co-inhibitory or co-stimulatory targeting antibodies can significantly improve the objective response rates and duration response. For example, a phase II trial combining nivolumab and ipilimumab demonstrated that the objective response rate of the combination was increased by up to 61% [175]. However, combination therapies require carefully dose-escalation studies of each antibody to assess the danger of over-stimulating the immune system. Anti-PD-L1-based bsAbs may show synergistic effects, increase durable antitumor responses, and decrease the danger of over-stimulating the immune system in patients who would not benefit from monospecific antibodies. The reason is that bsAbs are expected to bind to co-inhibitory and co-stimulatory molecules on the surface of primed T cells in the lymph nodes, which then migrate/traffic to the tumor site carrying the bsAbs with them. Once within the TME, T cells carrying bsAbs are expected to be able to engage with PD-L1 on the surface of tumor cells. Alternatively, primed lymphocytes may have already infiltrated the TME (so-called TILs). Thus, bsAbs may bind to co-inhibitory and co-stimulatory molecules on the surface of TILs, linking the TILs to tumor cells within the TME. These linkages are expected to enable TILs to show increased tumor infiltration and anti-tumor efficacy, thus provide exciting opportunities for anti-PD-L1-based therapy.

In the past six years, enormous strides have been made in the field of bsAb for cancer immunotherapy. There are at least 11 bsAbs targeting CD19 × CD3, EGFR × MET, gp100 × CD3, CD20 × CD3, BCMA × CD3, CTLA-4 × PD-1, CD20 × CD3, and GPRC5D × CD3, which have been approved for treatment of various cancers. In parallel with these approvals, over 50 anti-PD-L1-based bsAbs have been investigated in biological testing or in clinical trials since 2017. At least eleven proteins such as CTLA-4, LAG-3, PD-1, PD-L2, TIM-3, TIGIT, CD28, CD27, OX40, CD137 and ICOS are involved in these investigations. Twenty-two anti-PD-L1-based bsAbs are being evaluated to treat various advanced cancers in clinical trials, wherein the indications include NSCLC, SNSCLC, SCLC, PDA, MBNHL, SCCHN, UC, EC, TNBC, CC, and some other malignancies (Table 3). Pre-clinical results indicated that most of the anti-PD-L1-based bsAbs resulted in enhanced T-cell activation at lower concentrations compared with each parental antibody and their combination in vitro. Clinical trial results suggested that most of the anti-PD-L1-based bsAbs were well-tolerated and showed promising antitumor efficacy in patients with various advanced solid tumors. Since the approved and investigational bsAbs have shown more significant adverse reactions compared to monospecific anti-PD-L1 antibodies (Table 1 and Table 2), anti-PD-L1-based bsAbs should be further optimized to avoid or reduce these adverse reactions. One alternative approach dealing with this challenge is to develop novel modulators, for example, non-IgG-like bsAbs, traditional small molecules, or oligonucleotide aptamers simultaneously binding to both PD-L1 and co-inhibitory (or co-stimulatory) molecules. The reason is that these modulators offer distinct advantages over IgG-like bsAb such as being easy to manufacture, high tissue penetration, low immunogenicity, and fewer immune-related adverse events. Furthermore, small molecule modulators have much shorter half-lives than IgG-like bsAb. As a result, the severe cytokine release syndrome would be alleviated by breaking the administration of small molecule modulators.

## Figures and Tables

**Figure 1 molecules-29-00454-f001:**
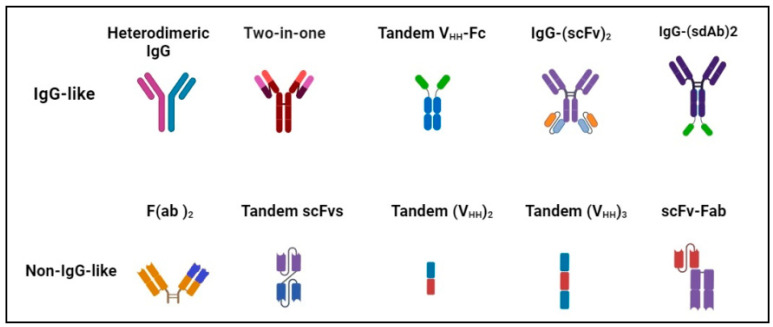
A collection of bsAbs formats. BsAbs are categorized into two main groups: IgG-like and non-IgGlike formats.

**Figure 2 molecules-29-00454-f002:**
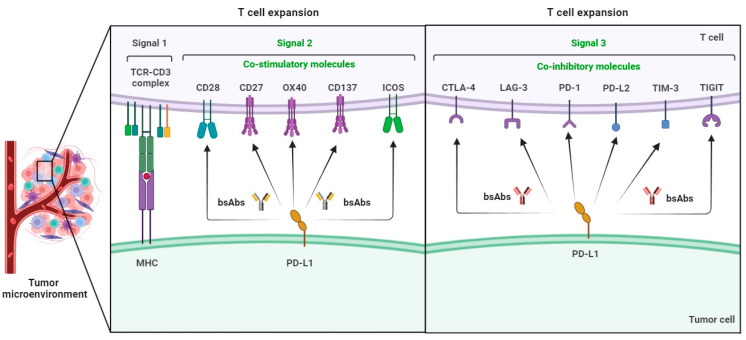
Co-inhibitory and co-stimulatory molecules involved in the development of anti-PD-L1-based bsAbs for cancer immunotherapy.

**Figure 3 molecules-29-00454-f003:**
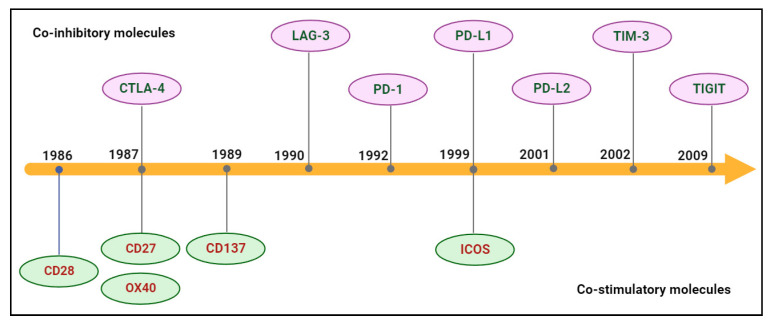
A brief timeline of the identification of co-inhibitory and co-stimulatory molecules involved in the development of anti-PD-L1-based bsAbs for cancer immunotherapy.

**Table 1 molecules-29-00454-t001:** Approved PD-L1 monospecific antibodies for cancer immunotherapy.

mAbs	Types	Approval	Investigators	Indications	Adverse Reactions
Atezolizumab	IgG1κ	2016 (US)	Genentech Roche	UC, NSCLC, SCLC, HCC, TNBC, Melanoma, Sarcoma	Fatigue, decreased appetite, nausea, urinary tract infection, pyrexia, constipation, dyspnea, cough, and MP
Durvalumab	IgG1κ	2017 (US)	AstraZeneca	UC, NSCLC, SCLC, HCC, BTC	Nausea, cough, fatigue, pneumonitis, upper respiratory tract infections, dyspnea, alopecia, constipation, decreased appetite, abdominal pain, rash, pyrexia, diarrhea, pruritis, and MP
Avelumab	IgG1λ	2017 (US)	EMD Serono Merck KGaA	MCC, UC, RCC	Fatigue, MP, diarrhea, nausea, infusion-related reaction, rash, decreased appetite, and peripheral edema
Envafolimab	Fc-fused	2021 (CN)	Alphamab	CC	NA
Sugemalimab	IgG4λ	2021 (CN)	CStone	NSCLC	NA
Adebrelimab	IgG4κ	2023 (CN)	Jiangsu Hengrui	SCLC	NA

UC, urothelial carcinoma; NSCLC, non-small-cell lung cancer; SCLC, small-cell lung cancer; HCC, hepatocellular carcinoma; TNBC, triple-negative breast cancer; BTC, biliary tract cancer; MCC, Merkel cell carcinoma; RCC, renal cell carcinoma; CC, cervical cancer; MP, musculoskeletal pain; NA, not available.

**Table 2 molecules-29-00454-t002:** Approved bsAbs as of the end of 2023.

mAbs	Mechanism	Approval	Investigators	Indications	Adverse Reactions
Blinatumomab	CD19 × CD3	2014 (US)	Amgen	ALL	Pyrexia, headache, peripheral edema, febrile neutropenia, nausea, hypokalemia, tremor, rash, and constipation
Emicizumab	FIXa × FX	2017 (US)	Genentech	HA	Injection site reactions, headache, and arthralgia
Amivantamab	EGFR × MET	2021 (US)	Janssen Biotech	NSCLC	Rash, IRR, paronychia, MP, dyspnea, nausea, fatigue, edema, stomatitis, cough, constipation, and vomiting; decreased lymphocytes, decreased albumin, decreased phosphate, decreased potassium, increased alkaline phosphatase, increased glucose, increased gamma-glutamyl transferase, and decreased sodium
Ozoralizumab	TNFα × HSA	2022 (JP)	Sanofi	RA	NA
Tebentafusp	gp100 × CD3	2022 (US)	Immunocore	UM	CRS, rash, pyrexia, pruritus, fatigue, nausea, chills, abdominal pain, edema, hypotension, dry skin, headache and vomiting
Faricimab	gp100 × CD3	2022 (US)	Immunocore	UM	CRS, rash, pyrexia, pruritus, fatigue, nausea, chills, abdominal pain, edema, hypotension, dry skin, headache and vomiting; decreased lymphocyte count, increased creatinine, increased glucose, increased aspartate aminotransferase, increased alanine aminotransferase, decreased hemoglobin, and decreased phosphate
Mosunetuzumab	CD20 × CD3	2022 (US)	Genentech	FL	CRS, fatigue, rash, pyrexia, and headache; decreased lymphocyte count, decreased phosphate, increased glucose, decreased neutrophil count, increased uric acid, decreased white blood cell count, decreased hemoglobin, and decreased platelets
Teclistamab	BCMA × CD3	2022 (US)	Janssen Biotech	MM	Pyrexia, CRS, MP, injection site reaction, fatigue, upper respiratory tract infection, nausea, headache, pneumonia, and diarrhea; decreased lymphocytes, decreased neutrophils, decreased white blood cells, decreased hemoglobin, and decreased platelets
Cadonilimab	CTLA-4 × PD-1	2022 (CN)	Akeso	CC	NA
Elranatamab	BCMA × CD3	2023 (US)	Pfizer	MM	CRS, fatigue, injection site reaction, diarrhea, upper respiratory tract infection, MP, pneumonia, decreased appetite, rash, cough, nausea, and pyrexia
Glofitamab	CD20 × CD3	2023 (US)	Genentech	DLBCL, LBCL	CRS, MP, rash, and fatigue; lymphocyte count decreased, phosphate decreased, neutrophil count decreased, uric acid increased, and fibrinogen decreased
Epcoritamab	CD20 × CD3	2023 (US)	Genmab	DLBCL	CRS, fatigue, MP, injection site reactions, pyrexia, abdominal pain, nausea, and diarrhea
Talquetamab	GPRC5D × CD3	2023 (US)	Janssen Biotech	MM	Pyrexia, CRS, dysgeusia, nail disorder, MP, skin disorder, rash, fatigue, weight decreased, dry mouth, xerosis, dysphagia, upper respiratory tract infection, diarrhea, hypotension, and headache

ALL, acute lymphoblastic leukemia; HA, hemophilia A; NSCLC, non-small-cell lung cancer; RA, rheumatoid arthritis; UM, uveal melanoma; FL, follicular lymphoma; MM, multiple myeloma; CC, cervical cancer; DLBCL, diffuse large B-cell lymphoma; LBCL, large B-cell lymphoma; IRR, infusion-related reactions; CRS, cytokine release syndrome; MP, musculoskeletal pain; NA, not available.

**Table 3 molecules-29-00454-t003:** Anti-PD-L1-based bsAbs targeting both co-inhibitory and co-stimulatory molecules for cancer immunotherapy.

bsAbs	Indications	Investigators	Highest Phase	NCT Number
**CTLA-4 × PD-L1**				
BCP-84	Advanced Cancer	Nanjing Legend Biotech	Biological Testing	
BCP-85	Advanced Cancer	Nanjing Legend Biotech	Biological Testing	
AB-04	Advanced Cancer	Sichuan Kelun-Biotech	Biological Testing	
KN-046	NSCLC, PDA	Jiangsu Alphamab	Phase III	NCT06020352 NCT05001724 (Terminated in 2023)
PR-001573	Advanced Cancer	Harbour BioMed	Biological Testing	
**LAG-3 × PD-L1**				
FS-118	Advanced Cancer, SCCHN	F-Star Delta invoX Pharma	Phase I/II	NCT03440437
ABL-501	Advanced Cancer	ABL Bio	Phase I	NCT05101109
IBI-323	ALK-Rearranged NSCLC	Innovent Biologics	Phase II	NCT05296278 NCT04916119
mPDL1HCv1-E-sLAG3	Advanced Cancer	GenScript Biotech	Biological Testing	
W-3669	Advanced Cancer	WuXi Biologics	Biological Testing	
hz7F10-hzB6	Advanced Cancer	Mabwell Bioscience	Biological Testing	
PB-68	Advanced Cancer	Merus NV	Biological Testing	
**PD-1 × PD-L1**				
Hanmi bsAb	Advanced Cancer	Hanmi Pharm	Biological Testing	
LY-3434172	Advanced Cancer	Eli Lilly	Phase 1	NCT03936959 (Completed in 2021)
anti-PD-1 × PD-L1 bsAb	Advanced Cancer	Sunshine Guojian	Biological Testing	
609-Fab-PD-L1-IgG4	Advanced Cancer	Sunshine Guojian	Biological Testing	
**PD-L2 × PD-L1**				
IMGS-001	Advanced Cancer	ImmunoGenesis	Pre-clinical	
Bi-201	Advanced Cancer	Biotheus Inc	Biological Testing	
**TIM-3 × PD-L1**				
LY-3415244	Advanced Cancer	Eli Lilly Zymeworks	Phase I	NCT03752177 (Terminated in 2021)
**TIGIT × PD-L1**				
BTP-21	Advanced Cancer	Legend Biotech	Biological Testing	
HLX-301	Advanced Cancer, Lymphoma, NSCLC	Henlius Biotech	Phase I/II	NCT05390528 NCT05102214
HB-0036	NSCLC	Huaota Biopharm	Phase I/II	NCT05417321
TIGIT-Fc-93-VH6	Advanced Cancer	I-Mab Biopharma	Biological Testing	
P-O-T	Advanced Cancer	Jiangsu Hengrui	Biological Testing	
HuPL721-T2353-scFab	Advanced Cancer	Nanjing Sanhome	Biological Testing	
PM-1022	Advanced Cancer	Biotheus Inc	Phase I/II	NCT05867771
**CD28 × PD-L1**				
CX694	Advanced Cancer	Inhibrx	Biological Testing	
Ab-1	Advanced Cancer	Janux Therapeutics	Biological Testing	
XENP-36764	Advanced Cancer	Xencor	Biological Testing	
**CD27 × PD-L1**				
CDX-527	Advanced Cancer	Celldex Therapeutics	Phase I	NCT04440943 (Completed in 2023)
**OX40 × PD-L1**				
28A10 × 2E4	Advanced Cancer	Inhibrx	Biological Testing	
IBI-327	Advanced Cancer	Innovent Biologics	Pre-clinical	
MEDI-1109	Advanced Cancer	AstraZeneca MedImmune	Pre-clinical	
BS-813	Advanced Cancer	JN Biosciences	Biological Testing	
KN-052	Advanced Cancer	Jiangsu Alphamab	Phase I	NCT05309512
EMB-09	Advanced Cancer	EpimAb Biotherapeutics	Pre-clinical	
L52-2D7H232	Advanced Cancer	Anhui Anke Biotechnology	Biological Testing	
**CD137 × PD-L1**				
ES-101	Thoracic Tumors, NSCLC, SCLC	Inhibrx Elpiscience	Phase I Phase I/II	NCT04009460 (Terminated in 2022) NCT04841538 (Withdrawn in 2022)
MCLA-145	Advanced Cancer	Merus	Phase I	NCT03922204
ATG-101	Advanced Cancer, MBNHL	OriCell Antengene	Phase I	NCT05490043 NCT04986865
ABL-503	Advanced Cancer	ABL Bio I-Mab Biopharma	Phase I	NCT04762641
FS-222	Advanced Cancer	F-star Therapeutics invoX Pharma	Phase I	NCT04740424
S-095012	Advanced Cancer	Pieris Servier	Phase I/II	NCT05159388
AP-203	Advanced Cancer	AP Biosciences	Phase I/II	NCT05473156
4-1BB (20H4.9) × PD-L1	Advanced Cancer	Roche	Biological Testing	
GEN-1046	NSCLC, UC, SCCHN, UC, EC, TNBC, CC	BioNTech Genmab	Phase II	NCT04937153 NCT03917381 NCT05117242
LBL-024	Advanced Cancer	Nanjing Leads Biolabs	Phase I/II	NCT05170958
9EN-FM	Advanced Cancer	Jiangsu Hengrui	Biological Testing	
PM-1003	Advanced Cancer	Biotheus	Phase I/IIa	NCT05862831
QL-301	Advanced Cancer	QLSF Biotherapeutics	Pre-clinical	
MF6797 × MF7702	Advanced Cancer	Merus	Biological Testing	
BH-3120	Advanced Cancer	Hanmi Pharmaceutical	Pre-clinical	
EuPD-1 × 94 kvt LHC218	Advanced Cancer	Eutilex	Biological Testing	
PR-004270	Advanced Cancer	Harbour BioMed	Biological Testing	
HK-010	Advanced Cancer	Anhui Anke Biotechnology	Pre-clinical	
1923Ab18	Advanced Cancer	Novarock Biotherapeutics	Biological Testing	
**ICOS × PD-L1**				
KY-1055	Advanced Cancer	Kymab	Pre-clinical	

NSCLC, non-small-cell lung cancer; SNSCLC, squamous non-small-cell lung cancer; SCLC, small-cell lung cancer; PDA, pancreatic ductal adenocarcinoma; MBNHL, mature B-cell non-Hodgkin’s lymphomas; SCCHN, squamous cell carcinoma of head and neck; UC, urothelial carcinoma; EC, endometrial carcinoma; TNBC, triple-negative breast cancer; CC, cervical cancer.

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
