# Peer review of "Anti-PD-L1-Based Bispecific Antibodies Targeting Co-Inhibitory and Co-Stimulatory Molecules for Cancer Immunotherapy"

_molecules, 2024, doi:10.3390/molecules29020454_

Round 1

Reviewer 1 Report

Comments and Suggestions for Authors

Geng provides a comprehensive review on bispecific antibodies that targetPD-L1 with one arm. The review merits publication. However,  language correction is required and some points are not clear. See below. Moreover, in the conclusion section,  existing combination therapies of anti-PD-L1 antibodies with other targeting antibodies should be (briefly) discussed and compared to benefits (and associated problems) of  bispecifics. This point is only briefly addressed in line 300.

A scheme should be provided displaying the schematic outline of all bispecifics that are discussed .For example: Li 173-176 the bispecifc construct is very hard to understand without a figure.

Li 43 replace wide type by wild type

Li 50: receptors not receptor

Li 78: number is wrong there are at least 13 bsAbs approved and 7 for cancer therapy now Table 2 should also be corrected

Li 118: Space before citation missing

Li 120: replace provides by provide

Li 121: replace “the CD28” by” CD28”

Li 129 bsAb should read bsAbs

Li 139: as should read to

Li 141 BSAb should read bsAb

Li 142: sepcicifcally binds should read that specifically binding

Li 143: SEB not defined

Li 145: 10ng/kg not mg/KG??? please check with the journal format. Same for li 154.

Li 145: TGI not defined.

Li 149 and 151: replace BSAb by bsAb, li 151 replace binds by bind

Li 155: remove “uptaken and”

Li 165: abbreviations BIRC and ORR not explained.

Li 173: What means simultaneously in this context?

Li 199: QW abbreviation not defined.

Li 200: is the Fab part of FS-118 derived from Avelumab?

Li 212: DCs not defined

Li 242: triple digit accuracy of affinity does not make sense!

Li 266: Replace “exhibited to have” by “displayed”

Li 276: Replace engaged with by engaged with one of

Li 280:  remove comma at end of line

Thoughout the paper: Replace T-cell by T cell. Replace B.cell by B cell.

Li 322. Replace Anti by anti

Li 326: 40% identity TO WHAT?

Li 369 abbreviation TILS only explained later.

Li 373: Replaced suggested by suggesting

Li 376: Replace November by November and October by October.

Li 386: -29.6% of what?

Li 429: Please consistently use Phase or phase

Li 434: IgG1antibody. Space missing.

Li 436: mab should read mAb

Li 443: Remove “,” after 0036

Li 444: please consistently use tumor or tumour

Li 452-453, 474,602,777,809,836 (!),923,927,935,936,984,985: triple digit precise KD useless, shorten to two digits after the "."

Li 468: is linked to N terminus should read is linked to the N-terminus.

Li 470: linked to THE N-terminus.

Li 471: hole not Hole

Li 478: what is meant with anti-tumor rate?

Li 495: replace molecules by molecule

Li 507, 518, 559,590,597, 632,678, 710, 738,789,816, 849, 863, 898,921, 932,949,979: replace was by is

Li 524:AB-1 was tested in vitro in combination with anti-TROP2XCDR3 engager. Cyno studies were without the engager? If yes why?

Li 542: KDa should read kDa

Li 563: replace CD4+ cells by CD4+ T cells.

Li 566: cells-injected transgenic hCD27-expressing . Hard to understand, please rephrase

Li 577, 581 NKTs TRAFs, abbreviations not defined

Li 592: Transfection of xx CHO cells with 28A10x2E4

Li 596: anti-PD not Anti

Li 607, also 608 What does a “tumor suppression rate of 80%” mean?

Li 615 inconsistent writing of Fcgamma

Li 625: Why is it imported to mention induction of Ki67+ Tmem cells? Is their induction beneficial or or problematic?

Li 646 PBMC should read PBMCs

Li 654: replacs fabs-in-tandem by Fabs-In-Tandem

Li 685 The purity.... This statement requires better explanation or deletion.

Li 732: treatment OF

Li 765: Whyis it important to note FcRn binding?

Li 844: Cmax should be defined when fist used, not here!

Li 850: N-terminus

Li 852 add "respectively" after 0.07 nM,

Li 877 : 0734 nM (?)

Li 895: replace "which were" by "which was"

Li 902; Remove “,”

Li 928: please consistently use hours or h

Li 944: anti-tumor efficacy. Hat does the value mean?

Li 846: WBC not defined

Li 968: disulfide

Li 972: Replace "while" by "upon"

Li 976: space missing after responses.

Li 999 response rate or response ratio?

Li 1012: These are at least 7 bispecifics that have been approved. See https://www.fda.gov/drugs/news-events-human-drugs/bispecific-antibodies-area-research-and-clinical-applications

Comments on the Quality of English Language

see above

Reviewer 2 Report

Comments and Suggestions for Authors

The author comprehensively summarized the strategy, advance and challenges in PD-L1 targeting immunotherapy. The historical overview and outlook for combination therapy is informative to readers.

One comment.

Besides its function in binding with PD-1 via cell-cell interactions, recent advances indicated the PD-L1 plays intracellular regulation of inflammatory programs. The authors may cite PD-L1 regulates inflammatory programs in cancer cells (PMID: 32839551, PMID: 32929201), immune cells (PMID: 37949473), and regulates genomic stability in cancer cells (PMID: 32350394, PMID: 33627620).

Comments on the Quality of English Language

Fine

Reviewer 3 Report

Comments and Suggestions for Authors

This paper discusses about recent advances of anti-PD-L1 based bispecific antibodies for cancer immunotherapy. The review is very well written, clearly organized and original. Language is correct and references are accurate. Tables and figures are well described and gather the necessary information to show the principal idea of each figure and table. Personally, I think that the manuscript is suitable for publication.  Below some few suggestions that could improve this paper:

  1. In addition to the role in the immune checkpoint, recent studies show that PD-1/PD-L1 interaction promotes tumorigenesis via mTOR signalling pathway in a group of cancers (Clark CA et al. Tumor-Intrinsic PD-L1 Signals Regulate Cell Growth, Pathogenesis, and Autophagy in Ovarian Cancer and Melanoma. Cancer Res. 2016; Kwak G et al.  Programmed Cell Death Protein Ligand-1 Silencing with Polyethylenimine-Dermatan Sulfate Complex for Dual Inhibition of Melanoma Growth. ACS Nano. 2017). The authors might discuss the immune-independent role of PD-L1 in the Introduction (line 40) for a complete description of PD-L1 function.
  2. When discussing AP203 bispecific antibodies, please refer to Huang PL et al., A bispecific antibody AP203 targeting PD-L1 and CD137 exerts potent antitumor activity without toxicity. J Transl Med. 2023.
  3. The author properly mentioned the need of alternative approaches dealing with safety issues related to bsAbs, including the use of small molecules. At this regard a brief discussion on the use of oligonucleotide aptamers could be a nice addition, given the new recent strategies successfully combining aptamers and anti-PD-L1 Abs for cancer treatments (i.e. Camorani et al. Aptamer targeted therapy potentiates immune checkpoint blockade in triple negative breast cancer. J Exp Clin Cancer Res. 2020; Passariello M et al. Novel Human Bispecific Aptamer-Antibody Conjugates for Efficient Cancer Cell Killing. Cancers (Basel). 2019).

Round 2

Reviewer 1 Report

Comments and Suggestions for Authors

Upon revision the paper improved significantly and can be recommended for publication as soon as the following two tiny points are fixed:

In the newly introduced text in line 1043 7 marketed bispecifics for cancer therapy are mentioned, however I count 11 in table 2.

Other point in li 1043 and table 2:

Avimantamab targets EGFR and MET (NOT METR)

Author Response

Dear Editor,

Thank you very much for your letter with reviewer’s comments. We appreciate your and reviewer’s comments, questions, and suggestions. We have answered questions from reviewers and made modifications according to their suggestions. We now respectfully submit the revised manuscript for your consideration for publication in Molecules.

In the following, we address reviewer’s suggestions/comments point by point:

Reviewer #1

Comments and Suggestions for Authors

Upon revision the paper improved significantly and can be recommended for publication as soon as the following two tiny points are fixed:

In the newly introduced text in line 1043 7 marketed bispecifics for cancer therapy are mentioned, however I count 11 in table 2.

We thank the reviewer for this suggestion.

7 was replaced by 11, and the phrase of "by FDA" was removed from this sentence.

There are at least 11 bsAbs targeting CD19×CD3, EGFR×MET, gp100×CD3, CD20×CD3, BCMA×CD3, CTLA-4×PD-1, CD20×CD3, and GPRC5D×CD3 have been approved for treatment of various cancers.

Other point in li 1043 and table 2:

Avimantamab targets EGFR and MET (NOT METR)

The phrase of "METR" was replaced by "MET" in the line of 1043 and Table2.